# Oxygen respiration and polysaccharide degradation by a sulfate-reducing acidobacterium

**Stefan Dyksma** [1] ✉ **& Michael Pester** [1,2] ✉

Sulfate-reducing microorganisms represent a globally important link between the sulfur and carbon cycles. Recent metagenomic surveys expanded the diversity of microorganisms putatively involved in sulfate reduction underscoring our incomplete understanding of this functional guild. Here, we use genome-centric metatranscriptomics to study the energy metabolism of *Acidobacteriota* that carry genes for dissimilation of sulfur compounds in a long-term continuous culture running under alternating anoxic and oxic conditions. Differential gene expression analysis reveals the unique metabolic flexibility of a pectin-degrading acidobacterium to switch from sulfate to oxygen reduction when shifting from anoxic to oxic conditions. The combination of facultative anaerobiosis and polysaccharide degradation expands the metabolic versatility among sulfate-reducing microorganisms. Our results highlight that sulfate reduction and aerobic respiration are not mutually exclusive in the same organism, sulfate reducers can mineralize organic polymers, and anaerobic mineralization of complex organic matter is not necessarily a multi-step process involving different microbial guilds but can be bypassed by a single microbial species.

Microorganisms can gain energy from dissimilatory sulfate reduction, a process that drives the biogeochemical sulfur cycle[1,2]. This process is of global relevance because of its tight connection to other element cycles, such as the carbon cycle. For example, sulfate reduction accounts for one-third of organic matter mineralization in the global seabed[3,4]. In anoxic freshwater sediments and soils, sulfate-reducing microorganisms (SRM) drive a hidden sulfur cycle and have strong control on the production of the potent greenhouse gas methane[5]. The true diversity of SRM is still unknown[6], yet metagenome mining greatly expanded the diversity of phylogenetic groups possessing the necessary genes for sulfate reduction[7].

Bacteria of the phylum *Acidobacteriota* are globally abundant and ubiquitous in soils[8,9] and sediments[10–13]. Recently, an energy metabolism based on the dissimilation of sulfur compounds was proposed for members of the *Acidobacteriota* present in marine sediments[14],

peatlands[15], thawing permafrost[16] and contaminated mine soils[17]. However, fundamental aspects regarding their ecophysiology are still unknown. Metagenomics revealed the potential for a variety of dissimilatory sulfur compound transformations, including disproportionation of sulfur cycling intermediates[14], tetrathionate/thiosulfate reduction[14] and utilization of organosulfonates for sulfur respiration[15]. More widespread among different acidobacterial lineages is the genetic trait for sulfite reduction or sulfide oxidation using the Dsr pathway[7,14–17], which is performed by the dissimilatory sulfite reductase complex (DsrAB, DsrC, DsrMK and DsrJOP). All *dsrAB*-containing *Acidobacteriota* identified to date encode the reductive-type DsrAB, however, based on genome data alone it is difficult to predict in which direction the pathway operates[18]. A further metabolic capacity that is widespread among the sulfur compound-dissimilating *Acidobacteriota* is the potential to respire oxygen, as aerobic respiratory chains are widespread in their genomes[14–17]. This makes it even more

[1]Leibniz Institute DSMZ—German Collection of Microorganisms and Cell Cultures, Department of Microorganisms, Braunschweig, Germany. [2]Technical University of Braunschweig, Institute of Microbiology, Braunschweig, Germany. ✉e-mail: stefan.dyksma@dsmz.de; michael.pester@dsmz.de

challenging to assess the mode of sulfur compound transformations in these bacteria.

To disentangle the physiology of sulfur compound-dissimilating *Acidobacteriota*, we followed sulfur-cycling in a bioreactor that was consecutively exposed to oxic and anoxic cycles over a period of more than 200 days to mimic the fluctuating redox conditions these microorganisms encounter in their natural habitats. Genome-centric metatranscriptomics embedded into controlled bioreactor operation identified polysaccharide degradation for an acidobacterium that actively switched between sulfate reduction and oxygen respiration under anoxic and oxic conditions, respectively.

# Results

## Alternating oxic and anoxic conditions selected for SRM capable of dealing with prolonged oxic conditions

To get hold of sulfur compound-dissimilating *Acidobacteriota*, we selected an environment that is dominated by *Acidobacteriota*, shows active sulfur cycling, and is characterized by periods of oxic and anoxic conditions in combination with an acidic pH as a selective pressure that discriminates against the majority of cultured SRM. This was true for peat soil from an acidic fen (pH 4-5) with active sulfur cycling and periodic water table fluctuations[5,19], which served as inoculum for a long-term continuous culture. The culture medium, containing pectin as the major carbon and energy source ($0.5\,g\,l^{-1}$) and sulfate as an electron acceptor (1 mM), was supplied at pH 4.5 and a low dilution rate ($0.025\,d^{-1}$). The source microbiome was exposed to fluctuating oxygen regimes with a periodical switch between oxic conditions (50% air·$O_2$ saturation) for one week and anoxic/sulfate-reducing conditions for four weeks, which selected for an adapted microbial community (Fig. 1). The pattern of sulfate reduction and acetate accumulation in the anoxic period as well as sulfate production and acetate consumption in the oxic period appeared relatively stable after approximately 100 days of operation (Fig. 1A). Sulfate reduction rates (SRR) reached up to $104\,\mu mol\,l^{-1}\,day^{-1}$, which were comparable to in situ SRR at the sampling site ($0$–$340\,\mu mol\,l^{-1}\,day^{-1}$)[20] and to rates determined in microcosm experiments with peat from the same location ($13\,\mu mol\,l^{-1}\,day^{-1}$)[21]. After 100 days of operation, sulfate reduction always reached a steady state in the anoxic period, i.e., the SRR was limited by the supply with fresh medium. The accumulation of acetate was likely due to the limitation of electron acceptor under anoxic conditions as pectin is supplied in excess in the medium. Other organic acids such as formate, propionate, butyrate and lactate were not detected, which indicated that acetate was a major intermediate of pectin degradation in the adapted community. Active sulfate reduction was corroborated by the production of sulfide under anoxic conditions (Supplementary Fig. S1). To prevent toxicity effects, hydrogen sulfide ($H_2S$) as the dominating sulfide species at the operating pH 4.5 was constantly removed from the bioreactor by the controlled gas supply that maintained a stable oxic or anoxic condition. Interestingly, we observed sulfate production in oxic periods despite no supply of fresh medium to reduce the growth of fast-growing aerobes. This indicated that also sulfur species of intermediate oxidation state must have been produced during bioreactor operation, allowing microbial sulfur oxidation to sulfate (Fig. 1A).

Based on 16S rRNA gene amplicons, the microbial α- and β-diversity changed upon bioreactor setup but stabilized after 100 days of operation indicating the establishment of an enrichment culture adapted to the bioreactor conditions (Fig. 1B, C). In this enrichment, *Acidobacteriota* became the second most abundant phylum after the *Pseudomonadota* accounting for 29% of the 16S rRNA gene amplicons at day 211, which is comparable to the inoculum (38%) and the undisturbed peat soil community (35–61%)[15,22]. Phyla that harbour known SRM, such as the *Desulfobacterota* and *Bacillota* accounted for an average relative abundance of 2.0% and 6.3%, respectively (Fig. 1D).

## Acidobacterium MAG CO124 can switch between dissimilatory sulfate reduction and oxygen respiration

Putatively sulfur compound-dissimilating *Acidobacteriota* were targeted using a genome-centric metatranscriptomic approach at day 172 and day 185. After dereplication, 106 metagenome-assembled genomes (MAGs) were recovered of which 19 affiliated with the *Acidobacteriota*, spread in at least six families (Fig. 2 and Supplementary Data 1). MAGs affiliating with known SRM that were typically found in peat, e.g., *Syntrophobacteraceae*, *Desulfomonilaceae* and *Desulfosporosinus*[19,23,24] were identified as well, numerically dominated the SRM community (Supplementary Data 1), but were not the scope of this study. Four of the 19 *Acidobacteriota* MAGs (CO124, BA147, BO159 and BA46) contained genes for dissimilatory sulfite reduction (*dsrAB*, *dsrC* and *dsrMKJOP*) (Fig. 2). Of those, one MAG (CO124) further contained all genes of the canonical sulfate reduction pathway including the genes for sulfate activation (*sat*), adenosine 5′-phosphosulfate reduction (*aprAB* and *qmoABC*) and sulfite reduction (*dsr* genes). Time-resolved qPCR analyses revealed that MAG CO124 was enriched over time in the bioreactor reaching up to $5.5 \times 10^4$ cells per ml (Supplementary Fig. S2), which corresponded to a relative abundance of 0.1%. This agrees well with the slow growth typically observed for members of the *Acidobacteriota*[25]. Phylogenomic analysis placed MAG CO124 into the family *Bryobacteraceae* with *Ca*. Sulfopaludibacter MAG SbA3 as the closest relative (Fig. 2). *Ca*. Sulfopaludibacter MAG SbA3 was originally recovered from the same acidic fen[15] that was used to inoculate the bioreactor. Across its genome, it shared 79% average nucleotide identity and 77% amino acid identity to MAG CO124 indicating that both represent distinct species of the same genus[26,27]. Like other putatively sulfur compound-dissimilating *Acidobacteriota* from peat[15,16], soil[17] and marine sediments[14], MAG CO124 encoded DsrL that acts as a NAD(P)H:acceptor oxidoreductase for DsrAB[28,29] and DsrD, which is an allosteric activator of DsrAB[30]. DsrL type 1a is exclusively found in sulfur oxidizers[29], whereas the *Acidobacteriota,* including MAG CO124 encode the DsrL type 2[14,15], which is spread among SRM, sulfur oxidizers and sulfur-disproportionating bacteria[29]. It has been recently suggested that *dsrD* is an indicator for a reductive sulfur compound metabolism when found together with *dsrABC* and *dsrMK*[30] as it is common in *dsr*-containing *Acidobacteriota*[14,15,17] (Fig. 2). Although there is published evidence that Dsr pathway genes were transcriptionally active in fen *Acidobacteriota*[15], it is so far not clear if the pathway operated in the reductive or oxidative direction. It has been further speculated that the pathway could be used for both, sulfate reduction and sulfide oxidation[15]. The controlled environment of the bioreactor allowed us to specifically link transcriptional profiles to a defined condition where either sulfate is reduced (anoxic) or produced (oxic) (Fig. 1). Therefore, we sequenced metatranscriptomes from an oxic period (day 172) and an anoxic period (day 185) for differential gene expression analysis. All *Acidobacteriota* MAGs were active in the transcriptome under both, oxic and anoxic conditions (Fig. 2). Transcription of *dsrAB* genes was used as a marker for sulfite reduction/sulfide oxidation activity among the four *Acidobacteriota* MAGs CO124, BA147, BO159 and BA46. The transcriptomes revealed that MAG CO124 showed active transcription of its *dsrAB* genes almost exclusively under anoxic conditions. In contrast, MAGs BA147, BO159 and BA46 showed little to no transcription of *dsrAB* irrespective of the condition (Supplementary Fig. S3). The major fraction of the coding sequences in the genomes of MAG BA147, BO159 and BA46 was not active at all and only very few genes were differentially transcribed between oxic and anoxic conditions (Supplementary Fig. S3). On the other hand, numerous genes of MAG CO124 were significantly upregulated ($P$ value adjusted < 0.05) under anoxic conditions, including *dsrAB* (Fig. 3A, B). Under oxic conditions, a high number of genes ($n = 213$)

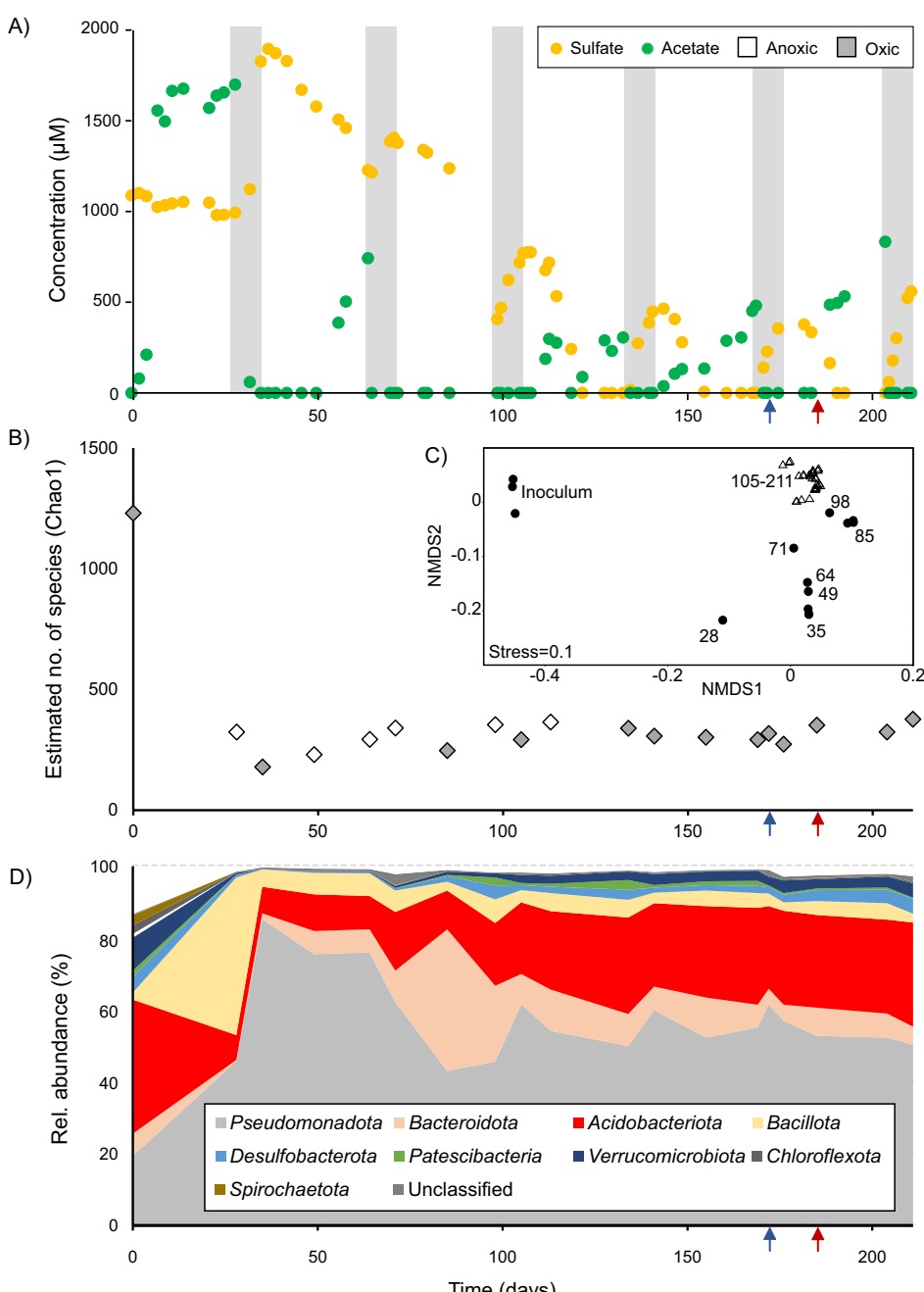

**Fig. 1 | Bioreactor performance and establishment of a stable enrichment culture over 211 days of operation.** Time-resolved concentration dynamics of sulfate and acetate (**A**). White and grey shaded areas represent anoxic (0% air-O$_2$ saturation) and oxic (50% air-O$_2$ saturation) conditions, respectively. Other organic acids such as malate, fumarate, succinate, citrate, lactate, propionate and butyrate were under the detection limit. Sulfate reduction rates (SRR) were calculated for periods with linear sulfate consumption. SRR were 104 µmol SO$_4^{2-}$ l$^{-1}$ day$^{-1}$ (day 113-122), 68 µmol SO$_4^{2-}$ l$^{-1}$ day$^{-1}$ (day 144–155) and 65 µmol SO$_4^{2-}$ l$^{-1}$ day$^{-1}$ (day 182-191).

Alpha diversity (**B**), beta diversity (**C**) and microbial community composition at phylum level (**D**) as determined by 16S rRNA gene amplicon sequencing. Samples for the 16S rRNA gene survey were sequenced in triplicates for most of the time points (filled symbols in **B**). Numbers within the NMDS plot (**C**) indicate the day of sampling. Samples for metagenome and -transcriptome sequencing were taken on days 172 (blue arrow) and 185 (red arrow). Source data are provided as Source Data file.

was upregulated as well (Fig. 3A), indicating a transcriptional response typical for facultative anaerobes.

To investigate this potential strategy of energy metabolism in MAG CO124, we compared the transcriptional profiles from the oxic and anoxic stage of the experiment in more detail. Under anoxic/sulfate-reducing conditions *sat*, *dsrC* and *aprAB* were among the top 50 most actively transcribed genes (average RPKM), thereby exceeding the activity of the DNA-directed RNA polymerase (*rpoA*) by up to 16-fold (Supplementary Data 2). Except *dsrJ*, all genes of the canonical

sulfate reduction pathway in MAG CO124 were significantly upregulated (*P* value adjusted <0.05) under anoxic conditions, while most of these genes were transcriptionally inactive under oxic conditions (Fig. 3B). Some *Desulfobacterota*, which are related to known SRM and encode a reductive type DsrAB, can operate the Dsr pathway in reverse for anaerobic sulfide oxidation[18,31]. However, such immediate re-oxidation of the produced sulfide can be excluded in the controlled environment of the bioreactor as electron acceptors such as nitrate and Fe(III) were not available. Congruent with the genomic repertoire

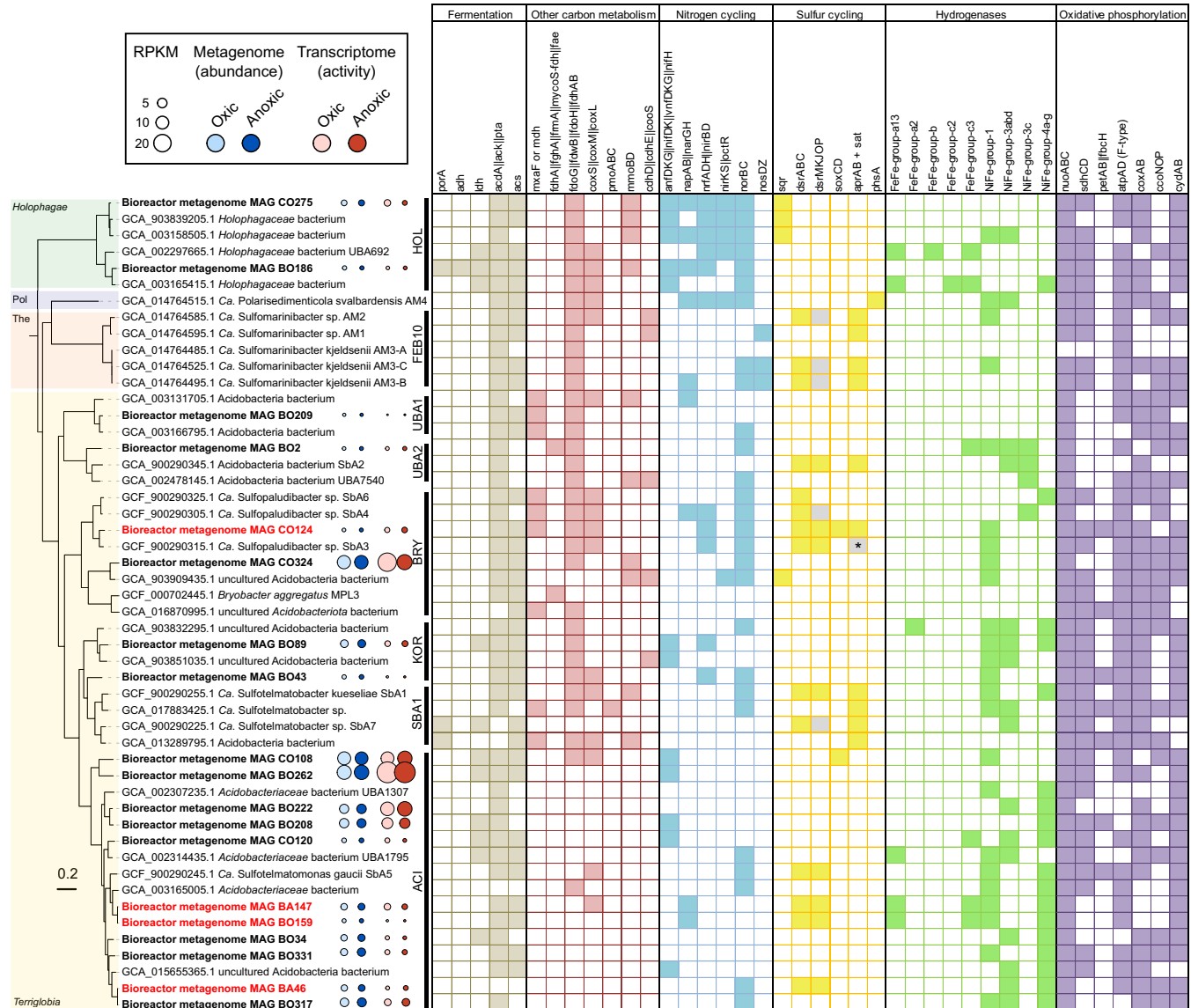

**Fig. 2 | Phylogenomic reconstruction and metabolic potential of 19 *Acidobacteriota* metagenome assembled genomes (MAGs) recovered in this study.** MAGs from this study (bold) that contained *dsrAB* are highlighted in red. The class is indicated at the left of the phylogenetic tree and family affiliation on the right (according to GTDB taxonomy[99]). Abbreviations are as follows: Pol, *Polarisedimenticola*; The, *Thermoanaerobaculia*; HOL, *Holophagaceae*; FEB10, uncultured family FEB10; UBA1, uncultured family UBA7541; UBA2, uncultured family UBA7540; BRY, *Bryobacteraceae*; KOR *Koribacteraceae*; SBA1, uncultured family SBA1; ACI, *Acidobacteriaceae*. The tree was constructed based on the concatenated alignment of single copy marker genes extracted with GTDB-Tk. Gene presence and absence was determined by an HMM-based annotation using METABOLIC-C. Presence is indicated by filled squares and gene absence by open squares. Grey squares indicate incomplete gene sets. The asterisk indicate gene presence reported in the manually curated annotation[15], but the complete genes were not detected using the HMM-based annotation. Gene abbreviations and functions are summarized in Supplementary Data 7. Normalized read and transcript abundance is given in reads per kilobase million (RPKM) and indicate the relative abundance in the metagenome and activity in the transcriptome, respectively. Source data are provided as Source Data file.

of the sulfur compound-dissimilating *Acidobacteriota* from peat[15], genes that encode sulfur-trafficking proteins commonly found in sulfur-reducing or disproportionating bacteria like TusA and DsrE[32] were not detected in MAG CO124. A sulfite dehydrogenase with low homology to SoxC and *c*-type cytochromes with low homology to SoxD and SoxX were encoded in MAG CO124, whereas other genes of the Sox multienzyme complex (*soxABYZ*) that are essential for thiosulfate oxidation[33,34] were absent. The role of the sulfite dehydrogenase remains unknown, but it was not active in the transcriptome, neither under oxic nor anoxic conditions (Supplementary Data 2). Other genes encoding sulfur compound metabolism enzymes, such as sulfide:quinone reductase (Sqr) and flavocytochrome c sulfide dehydrogenase (FccAB) were absent as well. Several

energy transducing enzyme complexes that are involved in dissimilatory sulfur compound metabolism interact with the quinone/quinol pool via a NrfD-like membrane protein. Such complexes include the dissimilatory sulfite reductase (DsrMKJOP), sulfur reductase (SreABC), dimethyl sulfoxide (DMSO) reductase (DmsABC), polysulfide reductase (PsrABC), tetrathionate reductase (TtrABC) and the sulfite-oxidizing enzyme SoeABC[35]. Moreover, NrfD homologs are present in enzymes other than those involved in sulfur compound metabolism, e.g., group 2 NiFe hydrogenase HydB, respiratory alternative complex ACIII and the quinone reductase complex QrcABCD that is often found in SRM[35,36]. Based on their phylogeny, NrfD-like proteins encoded in MAG CO124 were assigned to DsrMKJOP, QrcABCD, HydB and DmsC (Supplementary Fig. S4). The putative DMSO reductase (DmsABC) was

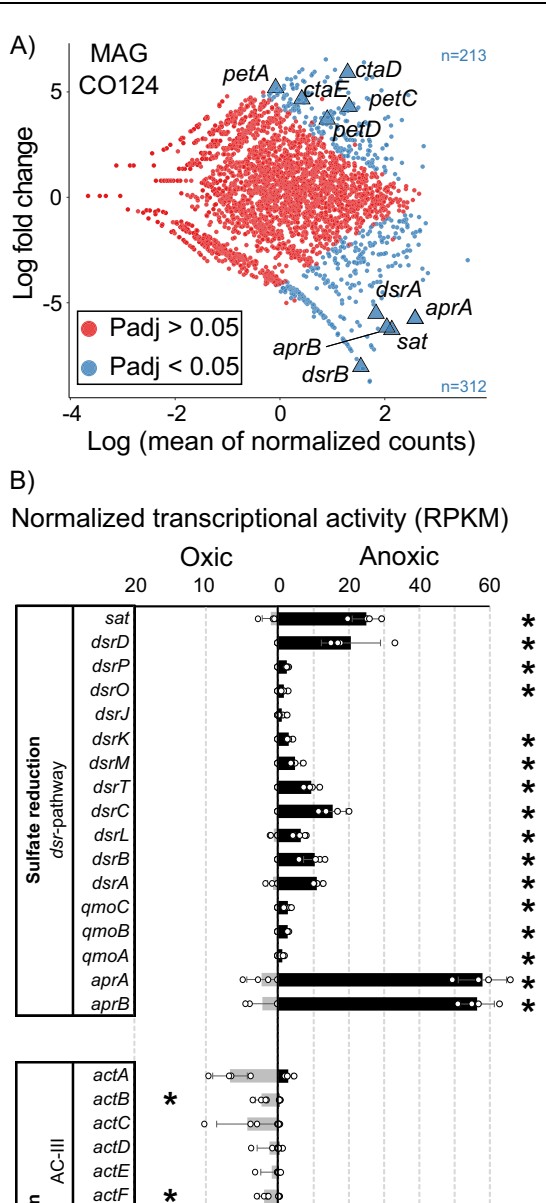

**Fig. 3 | Transcriptional activity of *Acidobacteriota* MAG CO124.** Differentially expressed genes between oxic and anoxic conditions (**A**). Each dot in the plot indicates a transcriptionally active gene in the genome. Genes that significantly changed in their transcriptional levels according to the Wald test as implemented in DESeq2 (*P* value adjusted < 0.05) were highlighted in blue. *P*-values were corrected for multiple testing (Benjamini-Hochberg). Selected hallmark genes of aerobic respiration (*ctaDE*, *petACD*) and sulfate reduction (*sat*, *aprAB* and *dsrAB*) are highlighted as triangles. Normalized transcriptional activity (RPKM) of Dsr pathway genes and genes encoding the aerobic respiratory chain (**B**). Bar charts show the mean ± SD of four technical replicates. Asterisks indicate genes that significantly changed (*P* value adjusted < 0.05) their transcriptional response between oxic and anoxic conditions. Significance testing was performed using the Wald test as implemented in DESeq2. *P*-values were corrected for multiple testing (Benjamini-Hochberg). Gene abbreviations and functions are shown in Supplementary Data 2 and 3. Source data are provided as Source Data file.

Currently, the class *Terriglobia* contains only three facultative anaerobic isolates[38,39], whereas the vast majority is described as strict aerobes. Accordingly, respiratory chains were identified in putatively sulfur compound-dissimilating *Terriglobia* from permanently oxic/hypoxic soil[17] and peat[15,16] as well as in the more distantly related *Acidobacteriota* from marine sediments (class *Thermoanaerobaculia* and *Polarisedimenticolia*)[14]. The terminal cytochrome oxidases encoded in these transcriptionally active *Acidobacteriota* were suggested to be used for oxygen defence[14,17] rather than aerobic growth. However, their ability to use aerobic respiration as primary energy metabolism is still unresolved. Seventeen out of 19 *Acidobacteriota* MAGs identified in this study affiliated with the class *Terriglobia* (Fig. 2). The active, sulfate-reducing MAG CO124 further encoded a complete oxygen respiratory chain (Supplementary Data 3). Almost all genes encoding the respiratory complex I, the NADH:ubiquinone oxidoreductase, were identified in MAG CO124. The succinate dehydrogenase (complex II) and both types of ubiquinol:cytochrome *c* oxidoreductase (cytochrome *bc* complex and alternative complex ACIII, complex III) were identified as well (Supplementary Data 3). A low affinity *caa₃*-type cytochrome *c* oxidase and a cytochrome *bd*-type oxidase were identified in MAG CO124 as terminal oxidases (respiratory complex IV). Like other *Acidobacteriota*, CO124 lack a high-affinity *cbb₃*-type cytochrome *c* oxidase[40]. The *caa₃*-type terminal oxidase, cytochrome *bc* complex, and alternative complex III were clearly upregulated in response to oxic conditions, as cumulatively indicated by significant transcriptional responses of their individual subunits (Fig. 3B). In contrast, the cytochrome *bd*-type oxidase of MAG CO124 was upregulated under anoxic conditions. Terminal oxidases of the *bd*-type do not pump protons[41]. They rather use intracellular protons for water formation with electrons derived from the extracytoplasmic side thereby contributing to a proton motive force[42]. It has been reported that the expression of the cytochrome *bd*-type oxidase increased under low oxygen and other stress conditions[41]. Moreover, it has been shown that the cytochrome *bd*-type oxidase allowed bacteria described as strict anaerobes to thrive under low oxygen concentrations[43,44] and can play a crucial role in protection against oxidative stress[45,46]. The discriminative response in transcription of the terminal oxidases in MAG CO124 points towards a different physiological function of these enzymes. The use of the *bd*-type oxidase seems to be strain-dependent among aerobic *Acidobacteriota*[40]. In MAG CO124, the *caa₃*-type cytochrome *c* oxidase likely contributed to respiratory activity under oxic conditions and the *bd*-type oxidase possibly contributed to other functions, such as response to oxidative stress under anoxic conditions.

### Acidobacterium MAG CO124 couples polysaccharide degradation to sulfate reduction and aerobic respiration

Pectin was the major carbon and energy source fed to the bioreactor. It is among the most complex macromolecules in nature, composed of various polysaccharides and it is abundant in the cell wall of terrestrial

not active in the transcriptome (Supplementary Data 2). The *qrcABCD* genes on the other hand showed transcription under anoxic conditions (Supplementary Data 2), which is consistent with the activity of this complex in SRM under sulfate-reducing conditions[36,37]. Taken together, this clearly points towards a functional Dsr pathway that operated in the reductive direction.

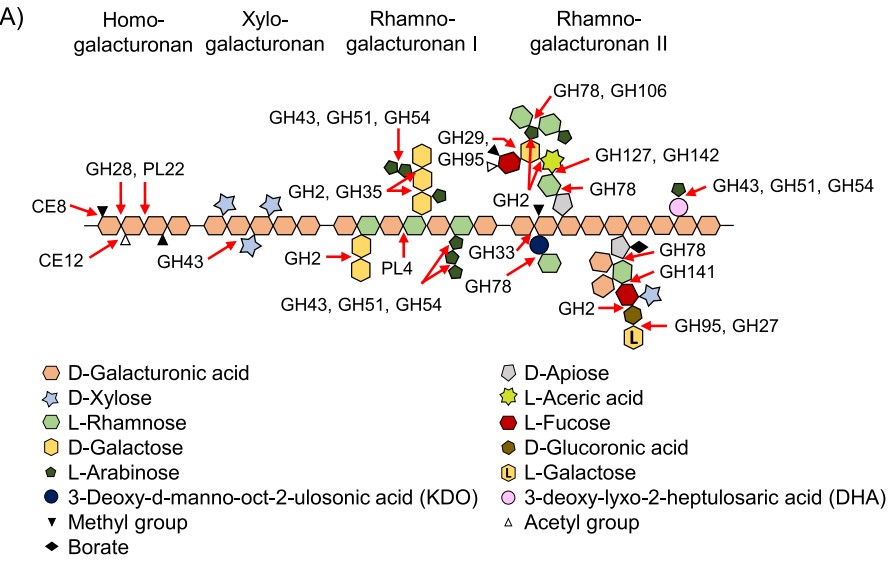

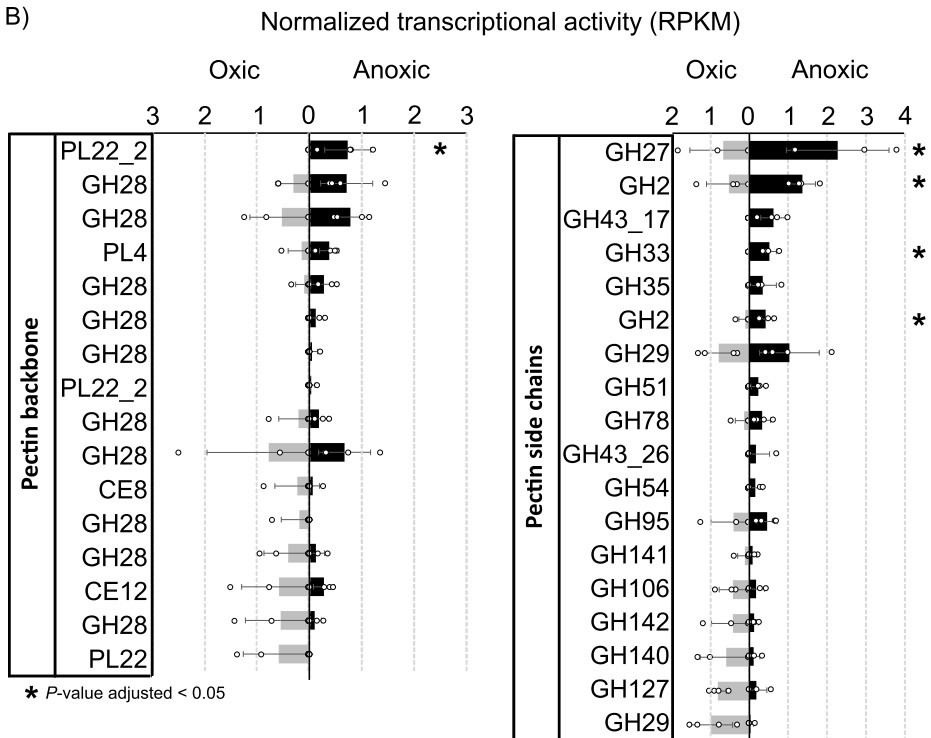

**Fig. 4 | Transcriptional pattern of central genes involved in pectin polysaccharide degradation of *Acidobacteriota* MAG CO124.** Structure of pectin polysaccharides as adapted after Mohnen (2008)[48], arrows indicate the potential target of the carbohydrate active enzymes encoded in MAG CO124 (**A**). Normalized transcriptional activity (RPKM) of glycoside hydrolases (GH), polysaccharide lyases (PL) and carbohydrate esterases (CE) that were potentially involved in the degradation of pectin polysaccharide is shown in **B**. Bar charts show the mean ± SD of four technical replicates. Asterisks indicate genes that significantly changed ($P$ value adjusted < 0.05) their transcriptional response between oxic and anoxic conditions. $P$-values were corrected for multiple testing (Benjamini-Hochberg). Source data are provided as Source Data file.

plants[47,48]. Thus, pectin can be an important source of carbon for microorganisms in terrestrial environments. However, the ability to degrade pectin was also found in marine bacteria, indicating a potential substrate niche for certain microorganisms in the ocean[49,50]. For the complete degradation of such a complex substrate the microorganisms require a diverse set of carbohydrate-active enzymes. The *Acidobacteriota* MAGs recovered in this study encoded numerous glycoside hydrolases (GH) and polysaccharide lyases (PL) that are generally involved in carbohydrate degradation (Supplementary Fig. S5). In fact,

the highest number of GH in all *Acidobacteriota* MAGs from the bioreactor were found in MAG CO124 ($n = 175$, genome size 7.29 Mb, 2.9% of all CDS). Pectin polysaccharides include, but are not restricted to, homogalacturonan, xylogalacturonan, rhamnogalacturonan I and rhamnogalacturonan II (Fig. 4A), which are among the most abundant constituents of pectin[47,48]. The transcriptional active enzymatic machinery of CO124 can extracellularly cleave methyl ester (CE8) and acetyl groups (CE12) of the pectin backbone, hydrolyse pectin side chains and degrade the galacturonan backbone[51-53] (Fig. 4A, B). Several

polygalacturonases of the family GH28 were encoded in the genome of MAG CO124 that were active in the transcriptome as well (Fig. 4B). Together with the oligogalacturonate lyases (PL22) and rhamnogalacturonate lyases (PL4) these enzymes likely initiate extracellularly the depolymerization of the pectin backbone. Most of the pectin-degrading carbohydrate-active enzymes in CO124 were transcribed under both, oxic and anoxic conditions, whereas an oligogalacturonan lyase (PL22) and some GHs that act on pectin side chains (GH2, GH27 and GH33) were upregulated specifically under anoxic/sulfate-reducing conditions (Fig. 4B). The simultaneous activity or even upregulation of pectin-specific carbohydrate-active enzymes (Fig. 4B) together with the Dsr pathway under anoxic conditions (Fig. 3B) strongly suggests polysaccharide degradation during sulfate reduction in MAG CO124. A transcriptionally active TonB-ExbB-ExbD complex might interact with TonB-dependent outer membrane receptors for the transport of cleaved oligosaccharides into the periplasm or they diffuse into the periplasm through porins[51,54] (Supplementary Data 4). A high number of acidobacterial orphan permeases were encoded in the genome of CO124 ($n = 102$), a substantial fraction of them were transcriptionally active ($n = 80$) and differentially expressed ($n = 13$) between oxic and anoxic conditions (Supplementary Data 4). This suggests an important role of these permeases in transport processes, which might include carbohydrate metabolism.

## Discussion

A sulfur cycle was fuelled by pectin as the major carbon substrate in the bioreactor (Fig. 1A). Here, we showed that MAG CO124 utilized pectin polysaccharides as a primary degrader in combination with sulfate reduction and aerobic respiration under anoxic and oxic conditions, respectively (Figs. 3 and 4). Known SRM usually use fermentation products such as alcohols (including sugar alcohols such as glycerol), organic acids and fatty acids while some can further grow with simple sugars like xylose, fructose and glucose or degrade hydrocarbons[1,2]. The transcriptome strongly supports the ability of MAG CO124 to use pectin during sulfate reduction, which expands the usable carbon and energy sources of SRM to polysaccharides as one of the major organic carbon pools on Earth. All so far known SRM are strict anaerobes, even though some can tolerate and reduce oxygen for a limited time period[55-60]. Interestingly, $O_2$-driven laboratory adaptive evolution enabled *Desulfovibrio vulgaris* Hildenborough not only to detoxify $O_2$ but also to gain energy from oxygen respiration under microoxic conditions (0.65% $O_2$)[61]. The differential expression profile of MAG CO124 strongly supports that a switch between an aerobic and sulfate-reducing energy metabolism is possible in SRM (Figs. 3 and 4).

In environments where related sulfur compound-dissimilating *Acidobacteriota* have been identified[14-17], this combination of metabolic traits might be key to cope with fluctuating conditions, e.g., during drying and rewetting in peatland soils, varying oxygen penetration during tidal cycles in coastal sediments and at oxic-anoxic transition zones in freshwater and marine sediments. These results expand our understanding of complex organic matter mineralization in anoxic environments, which is typically regarded as a three-step process under methanogenic (freshwater) conditions and as a two-step process under sulfate-reducing (marine) conditions[62]. Here, we provide evidence that the multi-step mineralization of complex organic matter to $CO_2$ can be bypassed in one step by sulfate-reducing *Acidobacteriota*, which are widespread in terrestrial and marine environments.

## Methods

### Sampling and bioreactor operation
Soil was sampled from the acidic fen "Schlöppnerbrunnen II" located in south-eastern Germany (50°08′38″N, 11°51′41″E). A detailed description of the fen was provided earlier[19]. Samples were taken from 10 to 20 cm depth of the peat layer and were transferred to sterile jars (one

liter volume). Samples were stored at 4 °C and used to inoculate the bioreactor after three days of storage using 20 ml of peat taken by a cut-off syringe. The reactor (two liter volume) was filled with 1000 ml medium composed of $KH_2PO_4$ 0.1 g l$^{-1}$, $(NH_4)_2SO_4$ 0.066 g l$^{-1}$, $MgSO_4 \times 7 H_2O$ 0.123 g l$^{-1}$, $CaCl_2 \times 2 H_2O$ 0.02 g l$^{-1}$, glucose monohydrate 0.1 g l$^{-1}$, pectin 0.5 g l$^{-1}$, trace element solution according to medium DSM141[63] (Leibniz Institute DSMZ-German Collection of Microorganisms and Cell Cultures), 1 ml l$^{-1}$, 20× vitamin solution according to DSM141[63] 0.5 ml l$^{-1}$ and MES hydrate 1.95 g l$^{-1}$. The bioreactor was initially operated as a batch culture without supply with fresh medium during the first 35 days. From that time point on, fresh medium was supplied only in the anoxic periods at a dilution rate of 0.025 day$^{-1}$ while the volume maintained constant using a level sensor. No additional medium was supplied in the oxic cultivation periods to reduce growth of fast-growing aerobes. The reactor was stirred at 15 rpm and operated at room temperature (20 °C ± 2 °C). Oxic or anoxic conditions were maintained by the controlled supply with compressed air or $N_2$ gas, respectively. The bioreactor was equipped with a pH electrode and the pH was maintained at 4.5 by manually dosing of 1 M HCl or 1 M NaOH solution.

### Analytical measurements
Inorganic anions and organic acids in the bioreactor were analyzed using ion chromatography. Anions (nitrite, nitrate, sulfate and phosphate) were separated in a SykroGel AX300 column (Sykam, Germany) using 4 mM $Na_2CO_3$ and 0.025 mM NaSCN as eluent. Organic acids (malate, fumarate, succinate, formate, citrate, lactate, acetate, propionate and butyrate) were separated in a SykroGel EX450 column (Sykam, Germany) using 7% acetonitrile and 0.7 mM perfluorbutanoic acid as eluent. Both, anions and organic acids were quantified using a conductivity detector. The sum of all sulfide species in the liquid bioreactor medium ($H_2S$, $HS^-$ and $S^{2-}$) as well as headspace hydrogen sulfide ($H_2S$) trapped in 2% (w:v) zinc acetate was quantified using the methylene blue method[64].

### Nucleic acid extraction and 16S rRNA gene amplicon sequencing
DNA for amplicon sequencing was extracted using the AllPrep PowerViral DNA/RNA kit (Qiagen, Netherlands) according to manufacturer's instructions. The microbial community composition in the bioreactor was determined by analyzing the hypervariable V4 region of the 16S rRNA gene using MiSeq (Illumina, CA, USA) sequencing of barcoded amplicons. Barcoded amplicons were prepared using primers 515 F (5′-GTGYCAGCMGCCGCGGTAA-3′) and 806 R (5′-GGAC-TACNVGGGTWTCTAAT-3′)[65-67] modified with a linker sequence. PCR conditions were as follows: initial denaturation, 95 °C for 3 min, followed by 10 cycles of 94 °C for 45 s, 50 °C for 60 s, 72 °C for 90 s and final elongation at 72 °C for 10 min. The Platinum™ Hot Start PCR master mix (Invitrogen, CA, USA) was used. PCR products were purified using AMPure XP beads (Beckman Coulter, CA, USA) and libraries were prepared using Nextera and TruSeq index adapter (Illumina) and the Platinum™ Hot Start PCR master mix (Invitrogen). Libraries were quantified with a Qubit 3.0 fluorimeter (Thermo Fisher Scientific, MA, USA) and paired end sequenced (2 × 300 bp) on the MiSeq platform using V3 chemistry (Illumina). Three replicated libraries were prepared for most of the sampled time points except for day 28, 49, 64, 71, 98 and 113. In total, 3.6 M reads were generated for downstream analysis.

Denoising, chimera filtering and amplicon sequence variant (ASV) delineation was performed using the DADA2 pipeline[68] (version 2022.2.0) implemented in Qiime2[69] (version 2022.2.1). Representative sequences were classified using the SINA classifier[70] and the SILVA SSU reference database 138.1[71]. Diversity measures were calculated in Qiime2 or PAST[72] (version 3.24) after random sub-sampling to the number of sequences of the smallest dataset ($n = 29,289$). NMDS analysis based on Bray-Curtis dissimilarity using ASVs from the sub-sampled dataset was performed with PAST.

### Metagenome and -transcriptome sequencing

Four replicates were sampled at day 172 (oxic period) and day 185 (anoxic period) for nucleic acid extraction. DNA for metagenome sequencing and RNA for transcriptome sequencing was extracted from each replicate using the AllPrep PowerViral DNA/RNA kit (Qiagen) according to manufacturer's instructions. For metagenome sequencing, DNA was sheared using a Covaris S220 (Covaris, MA, USA) resulting in fragment sizes of approximately 300–400 bp. Libraries were prepared using the NEBNext Ultra II DNA library prep kit (New England Biolabs, MA, USA) according to the manufacturer's protocol. For metatranscriptome sequencing, DNA was digested using the RNA Clean & Concentrator kit (Zymo Research, CA, USA). Total RNA was fluorometrically quantified using the RNA HS assay and a Qubit fluorometer (Thermo Fisher Scientific). Ribosomal RNA was depleted using the Ribo-off bacterial rRNA depletion kit (Vazyme Biotech, Nanjing, China). Total RNA and mRNA were checked with a Bioanalyzer instrument and the RNA 6000 Pico kit (Agilent Technologies, CA, USA). Libraries were prepared using Illumina's TruSeq stranded mRNA kit. Three metagenome libraries per time point and all four metatranscriptome libraries per time point were paired-end sequenced on an Illumina NextSeq 4000 platform using P3 reagents (2 × 150 bp).

### Genome-centric metagenomics

Raw metagenome reads were trimmed at quality score Q20 with a minimum length of 50 bp using bbduk.sh (BBTools version 38.22)[73]. All quality trimmed reads from the oxic period (day 172) were assembled together and all reads from the anoxic period (day 185) were assembled together using MEGAHIT (version 1.2.9)[74], resulting in two assemblies (Supplementary Data 5). To obtain average contig coverage, both read sets (oxic and anoxic) were mapped to the assemblies using Bowtie2 (version 2.3.5.1)[75] and coverage information was generated using the script jgi_summarize_bam_contig_depths included in MetaBAT2 (version 2.12.1)[76]. MetaBAT2 (version 2.12.1), MaxBin2 (version 2.2.7)[77] and MetaCoAG (version 1.0)[78] were used for differential coverage binning. Dereplication and aggregation was performed using DAS_Tool (version 1.1.4)[79], resulting in 161 metagenome assembled genomes (MAGs) from the oxic assembly and 163 MAGs from the anoxic assembly. The quality of the MAGs was assessed using CheckM (version 1.0.7)[80] and taxonomy was assigned using GTDB-Tk (version 2.1.1)[81]. MAGs from the oxic and anoxic assembly were further manually dereplicated based on completeness, contamination and taxonomy. This resulted in the final set ($n = 106$) of medium or high quality MAGs (≥50% completeness and ≤10% contamination or ≥90% completeness and ≤5% contamination, respectively)[82]. The four *dsrAB*-containing *Acidobacteriota* MAGs (CO124, BA147, BO159 and BA46) were further refined to obtain best possible quality. All metagenome reads were pooled and mapped to the MAGs using bbmap.sh (BBMap version 38.22) at a 99% identity cut-off and a local alignment with minimum overlap of 70%. Mapped reads were assembled using SPAdes (version 3.14.0)[83] and assembled contigs were binned using MetaBAT2. Finally, short contigs <5 kb were removed from the bins. To estimate the relative abundance of MAGs, RPKM (reads per kilobase million) tables were generated using bbmap.sh (99% identity cut-off and local alignment with minimum overlap of 70%) and by using the metagenomic OTU (mOTU) approach[84]. For the mOTU approach, protein coding genes were predicted from the assemblies and clustered at 95% global identity[85]. Downstream analysis focused on the universal single copy marker gene COG0202 (DNA-directed RNA polymerase).

All MAGs recovered in this study and additional *Acidobacteriota* MAGs (Fig. 2) were annotated using the METABOLIC-C pipeline[86] (version 4.0) and MetaErg[87] (version 1.2.0). Genes potentially involved in sulfur compound metabolism were further identified using DiSCo[88] (version 1.0.0). All protein coding sequences (CDS) of MAG CO124 were further analyzed using dbCAN2[89] and the CAZy database[90] to annotate carbohydrate active enzymes (Supplementary Data 6).

Protein translocation signal peptides[91] and transmembrane helices[92] were predicted for all CDS as well (Supplementary Data 6).

The concatenated alignment of single copy marker genes provided by the GTDB-Tk output was used for phylogenomic analysis. In total, 137 reference sequences and all 106 MAGs were selected for phylogenetic reconstruction with IQ-TREE 2 (version 2.2.0.3) using ultrafast bootstraps ($n = 1000$) after automatic substitution model selection[93]. For phylogenetic analysis of NrfD-like sequences, references were manually selected from GenBank[94] and the SWISS-PROT database[95]. The phylogenetic tree was reconstructed using 117 NrfD-like protein sequences with IQ-TREE 2 as described above.

### Genome-centric metatranscriptomics and differential gene expression analysis

Raw metatranscriptome reads were trimmed at quality score Q20 with a minimum length of 50 bp using bbduk.sh (BBTools version 38.22). Residual rRNA reads were removed from the quality trimmed sequences with RiboDetector[96] (version 0.2.6) using the option -e norrna. Non-rRNA reads were then mapped to the CDS of all MAGs using bbmap.sh (99% identity cut-off and local alignment with minimum overlap of 70%). The RPKM tables were used to follow the MAGs' transcriptional activity (Fig. 2). Raw transcriptome count tables from the anoxic period were corrected as the MAGs relative abundance slightly changed from day 172 (oxic period) to day 185 (anoxic period) (Supplementary Data 1). Raw counts from day 185 were multiplied by MAG coverage at day 172 divided by MAG coverage at day 185. The count tables were used as input for differential gene expression analysis using the DESeq2 (version 1.41.8) pipeline[97].

### Quantitative PCR (qPCR) analysis

Total 16 S rRNA genes of *Bacteria* and *Archaea* were quantified using the primer 1389 F (5′-TGTACACACCGCCCGT-3′) and 1492 R (5′-GGY-TACCTTGTTACGACTT-3′) with the LightCycler 480 SYBR Green I Master mix (Roche). The primer dsrB_453f (5′-TGCGGCTGGCCGTGGCTTGC-3′) and dsrB_705r (5′-TGCACTGCTCTTCGATCACT-3′) were designed to target the *dsrB* gene of MAG CO124. For the *dsrB*-targeted qPCR a standard curve from $10^2$ to $10^7$ copies was generated ($R^2 = 0.99$) using plasmid standards with the cloned *dsrB* gene of MAG CO124. Thermal cycling was carried as follows: initial denaturation at 95 °C for 5 min, followed by 40 cycles of denaturation at 95 °C for 30 sec, annealing at 65 °C (*dsrB* gene of MAG CO124) or 52 °C (total 16 S rRNA genes of *Bacteria* and *Archaea*) for 30 sec, and elongation at 72 °C for 30 sec, followed by melting curve analysis. Five ng DNA template was used for all qPCR reactions. The qPCR efficiency was 90-95%.

### Statistical analysis and reproducibility

Statistical analysis of differential gene expression was performed in R[98] employing the Wald test as implemented in DESeq2[97]. P values were corrected for multiple testing by the Benjamini-Hochberg approach as implemented in DESeq2. Adjusted P values < 0.05 were considered as statistically significant. No statistical method was used to predetermine sample size. Metagenomics and metatranscriptomics were performed in three and four technical replicates, respectively. 16S rRNA gene amplicon sequencing and qPCR analyses were performed in three technical replicates. The operating volume of the bioreactor (1 liter) was restricting the number of samples that could have been taken over time.

### Reporting summary

Further information on research design is available in the Nature Portfolio Reporting Summary linked to this article.

## Data availability

Amplicon sequences from the 16 S rRNA gene survey were deposited in NCBI BioProject PRJNA923133. Metagenomes, metatranscriptomes,

and MAGs are available under BioProject PRJNA923161. Source data are provided with this paper.

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

## Acknowledgements

We greatly acknowledge Petra Büsing for excellent technical support. We thank Gesa Ohlms for her help with the qPCR analysis, Petra Steffen for her help with R and David Kamanda Ngugi for fruitful scientific discussions. SD and MP were both financially supported by the Leibniz Institute DSMZ and the DFG (PE2147/3-1 to MP).

## Author contributions

The study was conceived and designed by M.P. Sampling of peat soil was done by M.P. Experiments were conducted by S.D. Metagenome and transcriptome data was analysed by S.D. S.D. wrote the manuscript and M.P. edited and approved the manuscript.

## Funding

## Competing interests

The authors declare no competing interests.
