## [Peer Review File · Nature Communications]

Oxygen respiration and polysaccharide degradation by a sulfate-reducing AcidobacteriumReviewer #1 (Remarks to the Author):

Based on bioreactor enrichment, Dyksma and Pester suggest that an uncultivated Acidobacterium is capable of switching between aerobic metabolism and dissimilatory sulfate reduction coupled to metabolizing the polymeric carbohydrate pectin. The existence of such a bacterium would shake up the so far accepted canonical role of sulfate-reducing microorganisms as degraders of simple fermentation products, and being strict anaerobes but not capable of utilizing polymeric organic matter directly.

Known was before that Acidobacteriota have the genomic potential for dissimilatory sulfate or sulfite reduction and oxygen respiration, and that the encoding genes are expressed; in one study (Hausmann et al. 2018) genes were found that point to a potential for using cellulose by as yet uncultivated Acidobacteriota with complete gene sets for dissimilatory sulfate reduction (or reduced sulfur compound oxidation). In the latter study, it was pointed out that it is difficult "to differentiate between reductive or oxidative sulfur metabolism solely based on (meta-)genome/transcriptome".

Comments

In summary, the authors present additional evidence for an Acidobacterium with the capability of both, sulfate reduction and aerobic respiration based on metagenomic and metatranscriptomic analysis of oxic and anoxic phases of the bioreactor. Since an absolute quantification of the novel Acidobacterium MAG CO124 is not available, as well as data comparing the abundance of the canonical sulfate-reducing bacteria, the quantitative importance of the novel Acidobacterium MAG CO124 in the bioreactor for sulfate reduction cannot be assessed. Transcriptomic data suggest expression of carbohydrate active enzymes in Acidobacterium MAG CO124, however, the role of the other carbohydrate active enzyme gene containing Acidobacteria (and other populations) was not revealed. A direct proof of carbohydrate assimilation by Acidobacterium MAG CO124 under sulfate reducing conditions is still open. Certainly, a pure culture is difficult to obtain, however, would allow to unequivocally prove polysaccharide utilization by a sulfate-reducing Acidobacterium.

1. Here, the authors used cycles of oxic and anoxic incubation for enriching their target microorganisms from the same peatland soil samples in a bioreactor fed with pectin (0.5 g L⁻¹) and sulfate (1 mM) and intermittent cycles of 50% air saturation. Sulfate seems to be stoichiometrically reduced to sulfide after some time (Sup Fig. 1), but reduced sulfur compound oxidizing activity appears to be present between oxic-anoxic cycles (Fig. 1); also acetate is formed. Since highly oxidizing conditions were present during oxic cycles, a multitude of oxidized sulfur compounds such as thiosulfate, tetrathionate or elemental sulfur could have been formed but were not tracked; these compounds can serve either as electron acceptors, or can be disproportionated.
2. Abundance of Acidobacteriota. The abundance was assessed based on relative 16S rRNA gene sequence abundances however, quantification of gene abundances is required for a solid estimation of Acidobacteriota abundance. Especially the absolute abundance of the population representing Acidobacterium MAG CO124 is of interest, since many other MAGs (18) of Acidobacteriota were found that do not have the genomic potential for sulfate reduction.
3. Abundance of metagenomic reads. The supposed sulfate-reducing Acidobacterium MAG CO124 had a coverage of 6-7% under oxic and anoxic conditions, whereas other Acidobacteria had coverages of >30 and up to >200 (Suppl. Tab. S1). Thus, it appears that Acidobacterium MAG CO124 was not highly abundant in the bioreactor. Some of the known sulfate-reducing microorganisms had coverages of >20 and up to >60 suggesting that their share in sulfate reduction activity cannot be neglected.
4. The presentation of the gene expression activity of glyco hydrolases and polysaccharide lyases is solely focused on Acidobacterium MAG CO124, but it is open how strongly other community members were expressing these gene sets; based on Supp. Fig. 4 there are at least other Acidobacteria that have large number of carbohydrate active enzymes. A direct proof of uptake of a carbohydrate by Acidobacterium MAG CO124 was not demonstrated but would unequivocally show that this Acidobacterium assimilates carbohydrates (and not other substrates, e.g., acetate present in the reactor fluid), while sulfate reduction conditions prevail.
5. The expression activity (and metagenome abundance) of Acidobacterium MAG CO124 appears low compared to that of other MAGs (Fig. 2, e.g. Bioreactor metagenome MAG CO108, Bioreactor metagenome MAG BO262, Bioreactor metagenome MAG CO324), all of which possess carbohydrate active enzymes as well, but do not contain sulfur cycling genes (except one has

soxCD). Thus, other Acidobacteria may have a more important role in carbohydrate metabolism in this bioreactor than Acidobacterium MAG CO124.

Reviewer #2 (Remarks to the Author):

- What are the noteworthy results?
- Will the work be of significance to the field and related fields? How does it compare to the established literature? If the work is not original, please provide relevant references.
- Does the work support the conclusions and claims, or is additional evidence needed?
- Are there any flaws in the data analysis, interpretation and conclusions? - Do these prohibit publication or require revision?
- Is the methodology sound? Does the work meet the expected standards in your field?
- Is there enough detail provided in the methods for the work to be reproduced?

The work by Dyksma and Pester has several novelty aspects that make this manuscript highly interesting for the readers, being definitely of sufficient ground-breaking character to ensure publication in Nat Comms. The utilization of polysaccharides (such as pectin) by sulfate reducers has not been so far proven and it is an important expansion on the knowledge of the organic matter degradation on anaerobic conditions. Linking the transcripts to known and shifting conditions to confirm the direction of the operating sulfur related pathways is also key to confirm the role of some Acidobacteria members in sulfate reduction. The methodology, data analysis and conclusions are sound. In my opinion, the manuscript is ready for publication after some minor changes.

A few comments/suggestions are the following:

1) On the dogma "sulfate reduction and aerobic respiration are mutually exclusive in the same organism" (line 47): I am not sure if this 'dogma' is applicable anymore as already a few studies challenged the previous view on SRM as strict anaerobes, showing how some *Desulfovibrio* spp. can temporarily respire oxygen, increasing resistance to oxic conditions (Mogensen et al. 2005; Santana 2008).

References:

Mogensen GL, Kjeldsen KU, Ingvorsen K (2005) *Desulfovibrio aerotolerans* sp. nov., an oxygen tolerant sulphate-reducing bacterium isolated from activated sludge. *Anaerobe* 11:339–349

Santana M (2008) Presence and expression of terminal oxygen reductases in strictly anaerobic sulfate-reducing bacteria isolated from salt-marsh sediments. *Anaerobe* 14:145–156

2) On literature novelty: there is some overlap with Li et al. (doi: 10.1038/s41522-022-00333-9). It would be helpful to better mention and clarify where this study went beyond (which does)...

3) On the metabolism and energy preservation: along the ms, several times I wondered until which point this microbe perform sulfate reduction and oxygen reduction linked to energy preservation and ultimately growth or not. E.g.: was oxygen-respiration true respiration or employed as oxygen detoxification (line 93), were genes linked to oxidative stress notorious in the aerobic phase? Did they benefit from oxygen respiration and grew from it (line 252)? Any data on this aspect? Did you measure increased biomass, or yield of nucleic acids per gram, protein, etc.? Any speculation over this to add to the discussion?

4) Regarding data availability: it is an increasing good practise to make available online the Rscripts used in each study pipeline for reproducibility. Consider to make them available in any online repository. I could not access the data on NCBI. Please make sure it is available prior to publication or at least immediately afterwards.

5) On the sulfide production mismatch with the sulfate reduced (as illustrated in Supplementary Figure 1 and regarding methods of sulfide measurements).

In the first phase the amount of reduced sulfate does not match the detected sulfide (line 199). Several questions arise here. From the simple why? Why in one phase and not in the other? Any chance to see a change in the microbial communities linked to this change of sulfide yield?

Anyway.. why?? Sulfate reduction does not stop in other compounds than ultimately sulfide. The sulfide resulting from sulfate reduction cannot be immediately oxidized in this phase as you do not

have other acceptors in the media... where does it go? Either you had oxygen diffusion that was immediately reduced by sulfide oxidation OR you are undercalculating the sulfide produced in this phase?. Was there possibly in average a lower pH in this phase before adjustment? (line 337) Do you have a log of the pH changes that the reactor was experiencing before readjusting it to pH 4.5? or a log of the amount added? A lower pH would have greatly influence the S-sulfide measured in the liquid phase resulting in underestimation. Another possibility thinking along could be that you underestimated the global sulfide production and in first phase was actually bigger and matching sulfate reduction and in phase 2 you had other sulfide coming from the reduction of other sulfur intermediates originated in the previous oxidative phase. Although this can sound strange, let's revisit the methodology. When you say: "as well as headspace hydrogen sulfide (H₂S) trapped in 2% (w:v) zinc acetate", (line 346): did you... A) Take sample of the headspace of the reactor and measure it? B) Place a solution of Zn Acetate in the off-line gas of the reactor and trap "all" the sulfide being degass from the reactor?

Opcion A)Methylene method will quantify the sulfide in the liquid, fine. But at pH 4.5, the majority of the sulfide is in the gas phase. If you are flushing continuously the reactor with nitrogen, you are flushing out most of the sulfide produced, and therefore, what you measure in the headspace is a big underestimation of the real amount that it has been produced. Opcion B) in my personal experience, when we grow isolates reducing a knowm/measured concentration of sulfate and therefore, we are 100% sure of the expected sulfide that we should measure off-line, we repetitively see big losses across the lines, saturation of the solution, etc. In summary, the sulfide measured in the Zn solution is often below the sulfide produced.

Other minor comments:

-line 40: you target the metabolism of Acidobacteriota in the omics analysis performed on the continuous culture, rather than in the culture itself. (semantic comment)

-line 43: It reads somehow long. Would rephrasing it like this work? To switch from sulfate to oxygen reduction under anoxic or oxic conditions, respectively coupled to the oxidation of pectin polysaccharides.

-line 77-78: maybe good to rephrase slightly the sentence so that it is more clear that (meta)genome mining is expanding the diversity of phylogenetic groups possessing the necessary genes for sulfate reduction and that ultimately could imply functionality. So make more clear that until proven (like in this paper), it is genetic potential.

-line 60: sulfur-dissimilatory can be interpreted as (elemental) sulfur while I assume that you want to refer to sulfur-compounds in general. If so, substitute. This observation can be extended to many other lines (such as 91, 108, 147, etc...). Another option is to clarify at the beginning that what you call sulfur-dissimilating microbes or reactions refer to a variety of sulfur compounds and not elemental sulfur exclusively.

-line 83: metagenomic revealed the genomic potential? (instead of capacity) of Acidobacteriota members.

-line 110: is not the argument of containing discriminatory selective pressure over culture SRM the same as the alternating oxic and anoxic conditions? If so, merged, if not, explain which other selective pressure you mean.

-Fig 1 panel B could be better supplementary material, instead it could be more interesting to show acidification/alkalinization of the system in response to the oxic/anoxic phases. For instance one could expect that in the oxic phase some sulfide or other S-intermediates would oxidize producing sulfuric acid and decrease pH while in the sulfate reduction phase, pH would increase. This further proves the on-going metabolic transformations. I recommend to measure redox for future studies too.

-line 125: what was the reason for a sulfate limitation phase, with no oxygen, no sulfate? Was there a scientific question behind it such as looking for a fermentative phase, or ..?

-line 127: your actinobacterial MAGS have all the pathways complete for oxidation of acetate?

-line 128: so far it fits with the known pathways until pyruvate and glyceraldehyde 3P, right? Want to comment on it?

-Supplementary figure 1: The Y axis is called amount of substance but maybe metabolites define better substrates and products of a reaction. I would suggest to express the sulfate in the real concentration being reduced instead of reduction, as it is really is. X axis better call Operation time (days).

-line 134: "sulfur species of intermediate oxidation state must have occurred in addition" I suggest to clarify this sentence as it could be misleading. Since so far sulfate reduction is known to go till sulfide without intermediate product accumulation, what you suggest it has to imply oxidation to

intermediate sulfur states of the produced sulfide. This might not be strictly clear how the sentence is written.

-line 153: 4 out of the 19 recovered MAGs (remembering the 19 is useful)

-line 159: It will be nice to better explain the implications in term of phylogeny that a threshold of 79% has. By the way, since the threshold of ANI for species definition is 95, did you try AAI for better check if they belong to different genus?

-all Acidobacteriota MAGs were active in the transcriptome under both, oxic and anoxic conditions. Since "In contrast, MAGs BA147, BO159 and BA46 showed little to no transcription of *dsrAB* irrespective of the condition", what were they doing? Short explanation, not deep detail.

-line 186: not very clear to me what you mean with this sentence "It was intriguing that a high number of genes (n=213) was upregulated with oxygen as well (Figure 3a)" do you mean that their function surprised you? Otherwise when you compare conditions, some are up, other are down, and it will depend on what you compare with. Since you just compare with each other, not with a control, it is not clear to me why it is surprising for you. Could you clarify this in the text?

-line 202: does the genome have rhodanases as found also important as well in S disproportionation or reduction? (e.g.: doi:10.1111/1462-2920.14442)

-line 230: meaning complete oxygen respiratory chain?

-line 273: since degradation of complex polysaccharides were not linked before to SRB, and not to other strict anaerobes such as methanogens, etc.. I am wondering, how novel is also the pectin degradation under strict anoxic conditions? Were the hydrolytic enzymes known to operate in the complete absence the oxygen? If no, maybe it is also nice to mention.

-line 297: xylose and sugar alcohols such as glycerol too.

-line 301: check better the oxygen reduction can tolerate for a limited time of period.

-line 328: correct the x of 20x into sign of fold instead of mathematical multiplication

-line 333: Since you do not refresh the medium in aerobic conditions (to reduce growth of fast growers), do you know at which pectin concentration this process was taking place? If you did not measure, maybe an approximation taking into account the stoichiometry of the sulfate reduction phase would help to elucidate the leftover amount with what the aerobic oxidation took place. This information is relevant to know the substrate concentration at which these microbes function.

-final remark; sorry for the delay. This time was on a review (I don't know if I will be review3, but #blame review3).

Reviewer #3 (Remarks to the Author):

The described study characterised a bioreactor fed with pectin and sulfate, seeded with material from an acidic fen. It was run for >200 days, alternative between an aerobic and anaerobic state. Based on a combination of 16S rRNA gene, metagenomic and metatranscriptomic approaches, metagenome assembled genomes were recovered and annotated. The study focuses on one of these, a novel Acidobacteria "CO124", which exhibited differential expression of several genes between aerobic and anaerobic bioreactor stages. Based on these analysis, they conclude this organism reduces sulphate in anaerobic conditions, using it as a terminal electron acceptor and degrading pectin. Overall I found the analyses compelling, with conclusions that are generally applicable. However, in the abstract and elsewhere it is claimed that it overturns 3 dogmas. The first two I agreed with, but the third, that "anaerobic mineralization of complex organic matter is not necessarily a multi-step process involving different microbial guilds but can be accomplished by one microorganism" is perhaps overstated. There are published examples of individual microorganisms being a part of multiple "guilds" within the same sample, so do not agree that it is a dogma to be overturned.

Given the metabolic reconstruction of the community, it appears that the authors conclusions linking individual genes to overall community functions (sulphate reduction, pectin degradation) are appropriate, but especially for the pectin degradation it remains possible (though not likely) that the genes are not doing the specific functions described. The organism is living in a complex

microbial community, and not all genes described have been extensively characterised. While it is apparent from the figure, the relative abundance of MAG CO124 in the community is comparatively low at the times when the MAG was recovered, and it is not clear how abundant it is according to the 16S-based analyses at other timepoints. The text should make mention of these numbers (if the 16S-based analyses are appropriate given challenges in linking amplicons to genomes). These data are important because if it is not the most abundant community member, then perhaps it is not contributing the most to community function, complicating interpretation. The authors should directly address this concern in the text.

I also had a small number of minor concerns:

396: Scaffold coverage was obtained by mapping with Bowtie2, but how was coverage derived from the mapping results?

P-values are reported for the metatranscriptomics, but these are derived from 2 conditions with 4 technical replicates each. It would be clearer for the reader if this was stated in the figure legend.

225: Ref 16 is a study of peat.

268: Most abundant constituents of what?

266 onwards: Reading the manuscript, it was unclear what analyses these conclusions are based on, at this stage in the manuscript. Also, since they are purely bioinformatic, they were quite definitive given the heterogeneity of functions (and simple lack of characterisation) found in many GH families. The conclusions should be stated less confidently.

Reviewer #1 (Remarks to the Author):

Based on bioreactor enrichment, Dykxma and Pester suggest that an uncultivated *Acidobacterium* is capable of switching between aerobic metabolism and dissimilatory sulfate reduction coupled to metabolizing the polymeric carbohydrate pectin. The existence of such a bacterium would shake up the so far accepted canonical role of sulfate-reducing microorganisms as degraders of simple fermentation products, and being strict anaerobes but not capable of utilizing polymeric organic matter directly.

Known was before that Acidobacteriota have the genomic potential for dissimilatory sulfate or sulfite reduction and oxygen respiration, and that the encoding genes are expressed; in one study (Hausmann et al. 2018) genes were found that point to a potential for using cellulose by as yet uncultivated Acidobacteriota with complete gene sets for dissimilatory sulfate reduction (or reduced sulfur compound oxidation). In the latter study, it was pointed out that it is difficult “to differentiate between reductive or oxidative sulfur metabolism solely based on (meta-)genome/transcriptome”.

Comments

In summary, the authors present additional evidence for an *Acidobacterium* with the capability of both, sulfate reduction and aerobic respiration based on metagenomic and metatranscriptomic analysis of oxic and anoxic phases of the bioreactor. Since an absolute quantification of the novel *Acidobacterium* MAG CO124 is not available, as well as data comparing the abundance of the canonical sulfate-reducing bacteria, the quantitative importance of the novel *Acidobacterium* MAG CO124 in the bioreactor for sulfate reduction cannot be assessed. Transcriptomic data suggest expression of carbohydrate active enzymes in *Acidobacterium* MAG CO124, however, the role of the other carbohydrate active enzyme gene containing *Acidobacteria* (and other populations) was not revealed. A direct proof of carbohydrate assimilation by *Acidobacterium* MAG CO124 under sulfate reducing conditions is still open. Certainly, a pure culture is difficult to obtain, however, would allow to unequivocally prove polysaccharide utilization by a sulfate-reducing *Acidobacterium*.

Response: Thank you for the valuable comments and suggestions that helped us to improve the manuscript. In the revised manuscript, we included two qPCR analyses to quantify total bacterial and archaeal 16S rRNA genes in comparison to *dsrB* genes of MAG CO124. For a PCR-independent confirmation of the newly obtained results, we also performed a mOTU (metagenomic OTU) analysis (doi 10.1038/s41467-019-08844-4) of our obtained MAGs as based on the single copy marker gene COG0202.

Furthermore, for a better understanding of the carbohydrate-active enzymes encoded in MAG CO124, we cultivated *Bryobacter aggregatus strain* MLP3, the closest related *Acidobacterium* that can utilize pectin (doi 10.1099/ijls.0.013250-0), with either pectin or glucose. We observed a clear upregulation of pectin-specific carbohydrate-active enzymes in the transcriptomes of *B. aggregatus* MLP3 when grown on pectin as compared to growth on glucose. This supports the suggested role of MAG CO124's glycoside hydrolases (e.g., GH28) and polysaccharide lyases (e.g., PL4, PL22; please see the additional figure provided for review purpose).

Please find below our point-by-point response.

1. Here, the authors used cycles of oxic and anoxic incubation for enriching their target microorganisms from the same peatland soil samples in a bioreactor fed with pectin (0.5 g L⁻¹) and sulfate (1 mM) and intermittent cycles of 50% air saturation. Sulfate seems to be stoichiometrically reduced to sulfide after some time (Sup Fig. 1), but reduced sulfur compound oxidizing activity appears to be present between oxic-anoxic cycles (Fig. 1); also acetate is formed. Since highly oxidizing conditions were present during oxic cycles, a multitude of oxidized sulfur compounds such

as thiosulfate, tetrathionate or elemental sulfur could have been formed but were not tracked; these compounds can serve either as electron acceptors, or can be disproportionated.

Response: We cannot exclude that partially oxidized sulfur compounds were disproportionated or used as electron acceptors by some members of the bioreactor community. However, MAG CO124 lack genes encoding tetrathionate reductase, thiosulfate reductase and sulfur carrier like TusA and DsrE, which can be found in sulfur disproportionating and sulfur reducing microorganisms (L 217-220; Supplementary Table S2 and S6). The metagenomes and metatranscriptomes provide no indication for other dissimilatory sulfur metabolism than sulfate reduction in MAG CO124. On the opposite, we detected a clear and significant transcriptional response of all genes encoding the full pathway of sulfate reduction to sulfide under anoxic conditions, including the activation of sulfate to APS and its reduction to sulfite. Genes encoding for the necessary enzymes (sat, aprAB) showed the highest transcript levels among all sulfate reduction pathway genes (Fig. 3B).

2. Abundance of Acidobacteriota. The abundance was assessed based on relative 16S rRNA gene sequence abundances however, quantification of gene abundances is required for a solid estimation of Acidobacteriota abundance. Especially the absolute abundance of the population representing Acidobacterium MAG CO124 is of interest, since many other MAGs (18) of Acidobacteriota were found that do not have the genomic potential for sulfate reduction.

Response: We followed your suggestion and designed a qPCR assay specific to the *dsrB* gene of MAG CO124. We compared these results to qPCR-quantified total bacterial and archaeal 16S rRNA genes to estimate the relative abundance of MAG CO124 (L 167 and new Supplementary Figure S2). Using this approach, we could show that CO124 was enriched over time to 5.5×10^4 cell per ml, which corresponds to 0.1 % of the total bacterial and archaeal community (new Supplementary Figure S2). As no DNA was left for qPCR analysis from the timepoints of metagenome/metatranscriptome sequencing (day 172 and day 185), we reanalyzed the metagenome data in order to get a precise estimation of the relative abundance of MAG CO124 using the metagenomic OTU (mOTU) approach. This approach confirmed a relative abundance of 0.1% for MAG CO124 (please see the new Supplementary Figure S2 and the revised Supplementary Table S1). We do not consider the low relative abundance as problematic because (i) our study was targeted towards the ecophysiology of Acidobacteria encoding the sulfate reduction pathway, (ii) we detected a clear increase in the population size of MAG CO124 in response to the applied conditions, (iii) MAG CO124 showed a clear and significant activation of the sulfate reduction pathway under anoxic conditions and of oxygen respiration under oxic conditions, and (iv) *Acidobacteriota* are well known for their very slow growth. In contrast, our study was not targeted towards *Acidobacteriota* dominating the process of sulfate reduction in the bioreactor and we do not make any claim in this direction throughout the manuscript.

3. Abundance of metagenomic reads. The supposed sulfate-reducing Acidobacterium MAG CO124 had a coverage of 6-7% under oxic and anoxic conditions, whereas other Acidobacteria had coverages of >30 and up to >200 (Suppl. Tab. S1). Thus, it appears that Acidobacterium MAG CO124 was not highly abundant in the bioreactor. Some of the known sulfate-reducing microorganisms had coverages of >20 and up to >60 suggesting that their share in sulfate reduction activity cannot be neglected.

Response: We followed your suggestion. To estimate the relative abundance of other SRM, we used the mOTU approach (see point 2 above). This Information has been included in the revised Supplementary Table S1. We also added a sentence in the Results section that highlights the numerical dominance of other SRM over MAG CO124 (revised manuscript L 161-162). Please see also our answer to point 2 above.

4. The presentation of the gene expression activity of glyco hydrolases and polysaccharide lyases is solely focused on *Acidobacterium* MAG CO124, but it is open how strongly other community members were expressing these gene sets; based on Supp. Fig. 4 there are at least other *Acidobacteria* that have large number of carbohydrate active enzymes. A direct proof of uptake of a carbohydrate by *Acidobacterium* MAG CO124 was not demonstrated but would unequivocally show that this *Acidobacterium* assimilates carbohydrates (and not other substrates, e.g., acetate present in the reactor fluid), while sulfate reduction conditions prevail.

Response: The scope of our study was to understand the ecophysiology of *Acidobacteriota* encoding the canonical sulfate reduction pathway. MAG CO124 is the only MAG from the bioreactor that transcribed both, the canonical sulfate reduction pathway and carbohydrate active enzymes specific for pectin degradation. This is one of the major novelties we intend to highlight in our study. We completely agree with reviewer 1, that other bacteria than MAG CO124 were very likely involved in/dominated the degradation of pectin polysaccharides in the bioreactor. This is also indicated by the large number of carbohydrate active enzymes in other *Acidobacteriota* MAGs (Supplementary Figure S5). As reviewer 1 already pointed out, those do not have the genomic potential for sulfate reduction and therefore their ecophysiology was beyond the scope of our study.

The point raised by reviewer 1 triggered us to conduct an additional experiment to validate our conclusions: Some of the glycoside hydrolases (GH28) and polysaccharide lyases (PL4, PL22) are well characterized and have specific functions in the depolymerization of pectin polysaccharides (further information and respective references can be found in the CAZy database, doi 10.1093/nar/gkt1178). Nevertheless, to validate the polysaccharide degradation potential of MAG CO124, we performed the following experiment. We cultivated *Bryobacter aggregatus*, the closest cultured relative that has been described in detail and is able to degrade pectin (doi 10.1099/ijs.0.013250-0), either with pectin or with glucose as sole substrate under oxic conditions. In *B. aggregatus*, pectin-specific carbohydrate active enzymes of the families GH28, PL22, CE19 were clearly transcriptionally upregulated or only transcriptionally activated when grown on pectin as compared to growth on glucose (please see the additional figure for review purpose below). This compares well to the transcriptional profile of MAG CO124 (Fig. 4, please note *B. aggregatus* encodes less glycoside hydrolases and polysaccharide lyases) and strongly supports the suggested role of the carbohydrate active enzymes transcribed or even upregulated in MAG CO124 under anoxic/sulfate-reducing conditions in our bioreactor setup.

Fig. Normalized transcriptional activity (RPKM) of *Bryobacter aggregatus* that was cultivated either with pectin or with glucose under oxic conditions. Four replicates of each condition were sequenced. The error bars represent the standard deviation of the four replicates. Coding sequences were annotated using MetaERG and EggNOG. Carbohydrate active enzymes were further predicted using dbCAN and the CAZy database. Carbohydrate active enzymes that were predicted by dbCAN with at least two approaches were presented as ‘High confidence’, while those that only have an annotation from MetaERG and/or EggNOG were presented as ‘Putative’. Transcriptional activity of the housekeeping gene DNA gyrase is depicted for comparison.

5. The expression activity (and metagenome abundance) of *Acidobacterium* MAG CO124 appears low compared to that of other MAGs (Fig. 2, e.g. Bioreactor metagenome MAG CO108, Bioreactor metagenome MAG BO262, Bioreactor metagenome MAG CO324), all of which possess carbohydrate active enzymes as well, but do not contain sulfur cycling genes (except one has soxCD). Thus, other *Acidobacteria* may have a more important role in carbohydrate metabolism in this bioreactor than *Acidobacterium* MAG CO124.

Response: As mentioned above the scope of our study was to understand the ecophysiology of a sulfate reduction pathway-encoding member of the *Acidobacteriota* rather than identifying the major pectin-degrading members in the bioreactor. Please see also our detailed responses above.

Reviewer #2 (Remarks to the Author):

The work by Dyksma and Pester has several novelty aspects that make this manuscript highly interesting for the readers, being definitely of sufficient ground-breaking character to ensure publication in Nat Comms. The utilization of polysaccharides (such as pectin) by sulfate reducers has not been so far proven and it is an important expansion on the knowledge of the organic matter degradation on anaerobic conditions. Linking the transcripts to known and shifting conditions to confirm the direction of the operating sulfur related pathways is also key to confirm the role of some *Acidobacteria* members in sulfate reduction. The methodology, data analysis and conclusions are

sound. In my opinion, the manuscript is ready for publication after some minor changes. A few comments/suggestions are the following:

Response: Thank you for reviewing our manuscript and for your positive feedback. We very much appreciate your valuable comments and suggestions that helped us to improve the manuscript. Please find below our point-by-point response.

1) On the dogma “sulfate reduction and aerobic respiration are mutually exclusive in the same organism” (line 47): I am not sure if this ‘dogma’ is applicable anymore as already a few studies challenged the previous view on SRM as strict anaerobes, showing how some *Desulfovibrio* spp. can temporarily respire oxygen, increasing resistance to oxic conditions (Mogensen et al. 2005; Santana 2008).

References:

Mogensen GL, Kjeldsen KU, Ingvorsen K (2005) *Desulfovibrio aerotolerans* sp. nov., an oxygen tolerant sulphate-reducing bacterium isolated from activated sludge. *Anaerobe* 11:339–349
Santana M (2008) Presence and expression of terminal oxygen reductases in strictly anaerobic sulfate-reducing bacteria isolated from salt-marsh sediments. *Anaerobe* 14:145–156

Response: We agree. Parts of the abstract have been rephrased more carefully. We cited the two studies mentioned by reviewer 2 and also included two more recent studies in the discussion that question the view on SRM as strict anaerobes (doi 10.1111/1462-2920.14466 and doi 10.3389/fmicb.2018.03159). The role of terminal oxidases (in particular cytochrome bd-type oxidases) for oxygen reduction and detoxification in anaerobic bacteria has been addressed in several studies. Among those, the studies from Mogensen *et al.* (2005) and Santana (2008) are good examples that highlight the diverse O₂-defense strategies in SRM. However, evidence for oxygen reduction coupled to energy conservation and ultimately growth is scarce. In a recent study, a strain of *Desulfovibrio vulgaris* Hildenborough was exposed to O₂-driven laboratory adaptive evolution and acquired via point mutations as well as deletions/insertions the ability to gain energy from oxygen respiration under microoxic conditions (0.65% O₂, doi 10.1111/1462-2920.14466). Since the enzymatic systems required for both sulfate and oxygen respiration were already present in the genome of *D. vulgaris*, only a limited number of mutations were apparently required to redirect the flow of reducing equivalents towards aerobic respiration coupled to growth (doi 10.1111/1462-2920.14466). We considered this study in the discussion of our revised manuscript (L 323).

2) On literature novelty: there is some overlap with Li et al. (doi: 10.1038/s41522-022-00333-9). It would be helpful to better mention and clarify where this study went beyond (which does)...

Response: Thank you for pointing this out. We included an additional sentence (L 244) to make this clearer. A major difference is that Li *et al.* (doi: 10.1038/s41522-022-00333-9) assumed that the terminal cytochrome oxidases encoded in the *Acidobacteriota* from permanently oxic/hypoxic soil are only used for the defense against oxygen.

3) On the metabolism and energy preservation: along the ms, several times I wondered until which point this microbe perform sulfate reduction and oxygen reduction linked to energy preservation and ultimately growth or not. E.g.: was oxygen-respiration true respiration or employed as oxygen detoxification (line 93), were genes linked to oxidative stress notorious in the aerobic phase? Did they benefit from oxygen respiration and grew from it (line 252)? Any data on this aspect? Did you measure increased biomass, or yield of nucleic acids per gram, protein, etc.? Any speculation over this to add to the discussion?

Response: According to your suggestion, we designed a qPCR assay specific to the *dsrB* gene of MAG CO124. We further quantified bacterial 16S rRNA genes to estimate the relative abundance of MAG CO124. The qPCR analysis has been included in the revised manuscript (L 167-169 and new Supplementary Figure S2). Using the specific qPCR assay that target the *dsrB* gene of MAG CO124 we could show that CO124 was enriched over time, reaching up to 5.5×10^4 cell per ml (new Supplementary Figure S2). The qPCR analysis also revealed a steep increase of *dsrB* gene copies of MAG CO124 from 1.3×10^4 per ml at day 169 (just before the oxic period) to 4.7×10^4 copies per ml at day 176 (after the oxic period). This substantial increase over the oxic period strongly indicate aerobic growth (new Supplementary Figure S2). The upregulated expression of genes encoding the oxygen respiratory chain under oxic conditions further points towards oxygen reduction linked to energy conservation (Figure 3, L 258-260). In contrast, genes encoding the bd-type cytochrome oxidase of MAG CO124 were upregulated under anoxic conditions being likely involved in residual oxygen detoxification under anoxic conditions (Figure 3, L 261-262). Terminal oxidases of the bd-type can be found in diverse strict anaerobic bacteria (including many SRM), which are implied to function in oxygen detoxification rather than aerobic growth (doi 10.1371/journal.pone.0123455).

4) Regarding data availability: it is an increasing good practise to make available online the Rscripts used in each study pipeline for reproducibility. Consider to make them available in any online repository. I could not access the data on NCBI. Please make sure it is available prior to publication or at least immediately afterwards.

Response: All 16S rRNA gene amplicon, metatranscriptome and metagenome data including the MAGs is already deposited in GenBank. The data will be released upon publication. The main analysis that was done with R was the differential analysis of RNA-Seq data using the DeSeq2 package. Here, we followed the workflow presented by Love *et al.* (doi 10.1186/s13059-014-0550-8) as cited in the manuscript.

5) On the sulfide production mismatch with the sulfate reduced (as illustrated in Supplementary Figure 1 and regarding methods of sulfide measurements).

In the first phase the amount of reduced sulfate does not match the detected sulfide (line 199). Several questions arise here. From the simple why? Why in one phase and not in the other? Any chance to see a change in the microbial communities linked to this change of sulfide yield? Anyway.. why?? Sulfate reduction does not stop in other compounds than ultimately sulfide. The sulfide resulting from sulfate reduction cannot be immediately oxidized in this phase as you do not have other acceptors in the media... where does it go? Either you had oxygen diffusion that was immediately reduced by sulfide oxidation OR you are undercalculating the sulfide produced in this phase?. Was there possibly in average a lower pH in this phase before adjustment? (line 337) Do you have a log of the pH changes that the reactor was experiencing before readjusting it to pH 4.5? or a log of the amount added? A lower pH would have greatly influence the S-sulfide measured in the liquid phase resulting in underestimation. Another possibility thinking along could be that you underestimated the global sulfide production and in first phase was actually bigger and matching sulfate reduction and in phase 2 you had other sulfide coming from the reduction of other sulfur intermediates originated in the previous oxidative phase. Although this can sound strange, let's revisit the methodology. When you say: "as well as headspace hydrogen sulfide (H₂S) trapped in 2% (w:v) zinc acetate", (line 346): did you... A) Take sample of the headspace of the reactor and measure it? B) Place a solution of Zn Acetate in the off-line gas of the reactor and trap "all" the sulfide being degass from the reactor?

Option A)Methylene method will quantify the sulfide in the liquid, fine. But at pH 4.5, the majority of

the sulfide is in the gas phase. If you are flushing continuously the reactor with nitrogen, you are flushing out most of the sulfide produced, and therefore, what you measure in the headspace is a big underestimation of the real amount that it has been produced. Opcion B) in my personal experience, when we grow isolates reducing a knowm/measured concentration of sulfate and therefore, we are 100% sure of the expected sulfide that we should measure off-line, we repetitively see big losses across the lines, saturation of the solution, etc. In summary, the sulfide measured in the Zn solution is often below the sulfide produced.

Response: Many thanks for pointing that out and for sharing your experiences with the methylene blue method. Although we can not fully exclude oxygen diffusion into the bioreactor, it seems unlikely that sulfide oxidation with possibly intruded oxygen traces would account for such a sizable fraction of the sulfide (Supplementary Figure S1). An underestimation of the produced sulfide seems to be more likely, as no other electron acceptors are available for the immediate anaerobic re-oxidation of the produced sulfide. However, it remains unclear why there was this pronounced difference between the two anoxic phases. The microbial community composition was stable over this time period (Figure 1B and C) as well as the concentration dynamics of sulfate and acetate (Figure 1A). The activity of intermediate sulfur compound disproportionating bacteria might have substantially contributed to sulfide production in the second phase. As you already pointed out, in this case we would have again underestimated the produced sulfide. Regarding our experimental setup, we placed a Zn-acetate solution in the gas outlet and assumed that we would trap all the sulfide being produced and degassed. Thanks for emphasizing that we can expect discrepancies in the sulfate/sulfide balance due to technical issues such as losses from the tubing or saturation of the solution. In the revised manuscript, we now refrain from any quantitative statement in respect to sulfide production and just mention that the observed sulfide production was in support of active sulfate reduction in the bioreactor (L 136-139).

Other minor comments:

-line 40: you target the metabolism of Acidobacteriota in the omics analysis performed on the continuous culture, rather than in the culture itself. (semantic comment)

Response: This sentence has been changed.

-line 43: It reads somehow long. Would rephrasing it like this work? To switch from sulfate to oxygen reduction under anoxic or oxic conditions, respectively coupled to the oxidation of pectin polysaccharides.

Response: Thanks. This has been changed accordingly.

-line 77-78: maybe good to rephrase slightly the sentence so that it is more clear that (meta)genome mining is expanding the diversity of phylogenetic groups possessing the necessary genes for sulfate reduction and that ultimately could imply functionality. So make more clear that until proven (like in this paper), it is genetic potential.

Response: Thanks for your suggestion. The sentence has been changed accordingly.

-line 60: sulfur-dissimilatory can be interpreted as (elemental) sulfur while I assume that you want to refer to sulfur-compounds in general. If so, substitute. This observation can be extended to many other lines (such as 91, 108, 147, etc...). Another option is to clarify at the beginning that what you

call sulfur-dissimilating microbes or reactions refer to a variety of sulfur compounds and not elemental sulfur exclusively.

Response: Thank you. To avoid misunderstanding by the reader, we now write sulfur compound-dissimilating to be more precise in our wording. This has been changed throughout the manuscript

-line 83: metagenomic revealed the genomic potential? (instead of capacity) of Acidobacteriota members.

Response: "Capacity" has been changed to "potential".

-line 110: is not the argument of containing discriminatory selective pressure over culture SRM the same as the alternating oxic and anoxic conditions? If so, merged, if not, explain which other selective pressure you mean.

Response: Indeed, thank you. This has been merged and extended by mentioning the acidic pH as an additional selective factor.

-Fig 1 panel B could be better supplementary material, instead it could be more interesting to show acidification/alkalinization of the system in response to the oxic/anoxic phases. For instance one could expect that in the oxic phase some sulfide or other S-intermediates would oxidize producing sulfuric acid and decrease pH while in the sulfate reduction phase, pH would increase. This further proves the on-going metabolic transformations. I recommend to measure redox for future studies too.

Response: Thank you, we will definitely follow your recommendations in future studies. As the pH was kept stable, we would like to keep Fig. 1B as is to provide evidence that the bioreactor community stabilized after ca. 100 days of operation, both at the level of alpha- and beta-diversity.

-line 125: what was the reason for a sulfate limitation phase, with no oxygen, no sulfate? Was there a scientific question behind it such as looking for a fermentative phase, or ..?

Response: The sulfate limitation phase was reached after more than 100 days of operation. As we were specifically interested in the *Acidobacteriota*, and the regular screening of 16S rRNA gene amplicons showed an enrichment of *Acidobacteriota* even in the anoxic periods under sulfate limitation (e.g., between day 113 and 133, or between day 155 and 169; Figure 1D), we decided not to change the experimental setup. As you noticed, this further allowed us to look into an anoxic phase where sulfate respiration requires efficient usage of sulfate delivered constantly in low amounts - a selective pressure that we hypothesized would select for sulfur compound-dissimilating *Acidobacteriota*.

-line 127: your actinobacterial MAGS have all the pathways complete for oxidation of acetate?

Response: A complete Wood-Ljungdahl pathway was not identified in any of the acidobacterial MAGs. However, other genes that can be involved in acetate oxidation such as acetyl-CoA synthase and citric acid cycle genes (including the glyoxylate bypath) are encoded in MAG CO124 and other acidobacterial MAGs (see also Figure 2 and Table S6).

-line 128: so far it fits with the known pathways until pyruvate and glyceraldehyde 3P, right? Want to comment on it?

Response: Correct, this is expected, but only for the homogalacturonan constituent of pectin. Other major polysaccharides of pectin are, e.g., rhamnogalacturonan I and rhamnogalacturonan II that can have long and variable side chains (see also Figure 4). Thus, several different fermentation products were described for isolated anaerobic pectinolytic bacteria, e.g., methanol, ethanol, formate, propionate, butyrate, and succinate (doi 10.1099/ijsem.0.002395, doi 10.1007/s00792-011-0399-7 and doi 10.1099/00221287-128-2-393).

Supplementary figure 1: The Y axis is called amount of substance but maybe metabolites define better substrates and products of a reaction. I would suggest to express the sulfate in the real concentration being reduced instead of reduction, as it is really is. X axis better call Operation time (days).

Response: Thank you. The labelling of the both axes has been changed accordingly. As for the sulfate, we did not fully understand what reviewer 2 meant here but interpreted it as stating sulfate consumption instead of sulfate reduction. This was changed accordingly.

-line 134: "sulfur species of intermediate oxidation state must have occurred in addition" I suggest to clarify this sentence as it could be misleading. Since so far sulfate reduction is known to go till sulfide without intermediate product accumulation, what you suggest it has to imply oxidation to intermediate sulfur states of the produced sulfide. This might not be strictly clear how the sentence is written.

Response: To avoid misunderstanding, we state now that sulfur species of intermediate oxidation state likely occurred during bioreactor operation without speculating on the processes that led to the production of these sulfur species (L 136-139).

-line 153: 4 out of the 19 recovered MAGs (remembering the 19 is useful)

Response: Done as suggested.

-line 159: It will be nice to better explain the implications in term of phylogeny that a threshold of 79% has. By the way, since the threshold of ANI for species definition is 95, did you try AAI for better check if they belong to different genus?

Response: Done as suggested (L 174-176).

-all Acidobacteriota MAGs were active in the transcriptome under both, oxic and anoxic conditions. Since "In contrast, MAGs BA147, BO159 and BA46 showed little to no transcription of *dsrAB* irrespective of the condition", what were they doing? Short explanation, not deep detail.

Response: These MAGs may have switched from aerobic respiration to a fermentative lifestyle. Under anoxic conditions, genes that are potentially involved in fermentation such as H₂-evolving hydrogenases, pyruvate:ferredoxin oxidoreductase and acetate kinase showed higher transcriptional activity in these MAGs. As we did not focus on a detailed analysis of their transcriptomes, we would rather prefer to omit any speculations about their functions in the bioreactor.

-line 186: not very clear to me what you mean with this sentence “It was intriguing that a high number of genes (n=213) was upregulated with oxygen as well (Figure 3a)” do you mean that their function surprised you? Otherwise when you compare conditions, some are up, other are down, and it will depend on what you compare with. Since you just compare with each other, not with a control, it is not clear to me why it is surprising for you. Could you clarify this in the text?

Response: We agree and just state our observation now (L 202-205).

-line 202: does the genome have rhodanases as found also important as well in S disproportionation or reduction? (e.g.: doi:10.1111/1462-2920.14442)

Response: Two rhodanese-like domains without a clear function were encoded in MAG CO124 (Supplementary Table S2). They share highest sequence similarity with rhodanese-like domain-containing proteins of *Bryobacter aggregatus* or *Paludibaculum fermentans*, two isolates that affiliate with the family *Bryobacteraceae*. Both species have been described in detail but no dissimilatory sulfur metabolism was reported (doi 10.1099/ijs.0.013250-0 and doi 10.1099/ijs.0.066175-0).

-line 230: meaning complete oxygen respiratory chain?

Response: Done as suggested.

-line 273: since degradation of complex polysaccharides were not linked before to SRB, and not to other strict anaerobes such as methanogens, etc.. I am wondering, how novel is also the pectin degradation under strict anoxic conditions? Were the hydrolytic enzymes known to operate in the complete absence the oxygen? If no, maybe it is also nice to mention.

Response: Thanks for pointing that out. Several anaerobic pectinolytic bacteria have been isolated and described (e.g., doi 10.1099/ijsem.0.002395, 10.1007/s00792-011-0399-7 and 10.1099/00221287-128-2-393). Such polysaccharide hydrolyzing/fermenting are for example associated with the human gut and the rumen of ruminants.

-line 297: xylose and sugar alcohols such as glycerol too.

Response: Thanks, this has been added.

-line 301: check better the oxygen reduction can tolerate for a limited time of period.

Response: We modified this section in the discussion and included recent literature (L 323-325).

-line 328: correct the x of 20x into sign of fold instead of mathematical multiplication

Response: Done as suggested.

-line 333: Since you do not refresh the medium in aerobic conditions (to reduce growth of fast growers), do you know at which pectin concentration this process was taking place? If you did not measure, maybe an approximation taking into account the stoichiometry of the sulfate reduction

phase would help to elucidate the leftover amount with what the aerobic oxidation took place. This information is relevant to know the substrate concentration at which these microbes function.

Response: We did not measure pectin concentrations at any timepoint. Only a very rough estimation would be possible based on the reduced sulfate and considering the accumulated acetate and assuming that pectin is solely composed of galacturonic acid. With these assumptions, approximately half of the supplied pectin is consumed at the end of the anoxic period.

-final remark; sorry for the delay. This time was on a review (I don't know if I will be review3, but #blame review3).

Reviewer #3 (Remarks to the Author):

The described study characterised a bioreactor fed with pectin and sulfate, seeded with material from an acidic fen. It was run for >200 days, alternative between an aerobic and anaerobic state. Based on a combination of 16S rRNA gene, metagenomic and metatranscriptomic approaches, metagenome assembled genomes were recovered and annotated. The study focuses on one of these, a novel Acidobacteria "CO124", which exhibited differential expression of several genes between aerobic and anaerobic bioreactor stages. Based on these analysis, they conclude this organism reduces sulphate in anaerobic conditions, using it as a terminal electron acceptor and degrading pectin. Overall I found the analyses compelling, with conclusions that are generally applicable. However, in the abstract and elsewhere it is claimed that it overturns 3 dogmas. The first two I agreed with, but the third, that "anaerobic mineralization of complex organic matter is not necessarily a multi-step process involving different microbial guilds but can be accomplished by one microorganism" is perhaps overstated. There are published examples of individual microorganisms being a part of multiple "guilds" within the same sample, so do not agree that it is a dogma to be overturned.

Response: We very much appreciate your constructive comments and suggestions. We addressed them by conducting additional experiments concerning the abundance and pectin-degradation capability of MAG CO124 (for details please see below). We rephrased the abstract and other related sections in refraining from the phrases like "...break three central dogmas..." and changing them to "...highlight that ... the anaerobic mineralization of complex organic matter is not necessarily a multi-step process involving different microbial guilds but can be accomplished by one microorganism." We hope reviewer 3 can agree with this re-phrasing.

Given the metabolic reconstruction of the community, it appears that the authors conclusions linking individual genes to overall community functions (sulphate reduction, pectin degradation) are appropriate, but especially for the pectin degradation it remains possible (though not likely) that the genes are not doing the specific functions described. The organism is living in a complex microbial community, and not all genes described have been extensively characterised. While it is apparent from the figure, the relative abundance of MAG CO124 in the community is comparatively low at the times when the MAG was recovered, and it is not clear how abundant it is according to the 16S-based analyses at other timepoints. The text should make mention of these numbers (if the 16-based analyses are appropriate given challenges in linking amplicons to genomes). These data are important because if it is not the most abundant community member, then perhaps it is not contributing the most to community function, complicating interpretation. The authors should directly address this concern in the text.

Response: According to your suggestion, we included a quantitative PCR (qPCR) analysis that targeted the *dsrB* gene of MAG CO124. Total 16S rRNA genes were quantified as well using qPCR to estimate the relative abundance of MAG CO124 at the other timepoints. Further, we included the metagenomic OTU (mOTU) approach (doi 10.1038/s41467-019-08844-4) to estimate the relative abundance of the MAGs at the timepoints of metagenome sequencing. MAG CO124 increased in absolute abundance over time reaching up to 5.5×10^4 cells per ml (new Supplementary Figure S2), as determined by the qPCR assay specific to the *dsrB* gene of MAG CO124, thereby accounting for a relative abundance of approximately 0.1%. We also addressed this in the text of the revised manuscript (L 167) to make it clearer that MAG CO124 is not among the most abundant community members. We are aware and agree that other bacteria than MAG CO124 very likely contributed to the degradation of pectin polysaccharides in the bioreactor, which is also indicated by the large number of carbohydrate active enzymes in other *Acidobacteriota* MAGs (Supplementary Figure S5). As those MAGs do not have the genomic potential for sulfate reduction they were not the target of our study. CO124 is the only MAG from the bioreactor that transcribed both, the canonical sulfate reduction pathway and carbohydrate active enzymes specific for pectin degradation. The rather low relative abundance of MAG CO124 (0.1%) is comparable to the abundance of the other SRM in the bioreactor community (0.04-0.4%; modified Supplementary Table S1), which indicates that overall sulfate-reducing activity is shared by these bacteria.

Regarding the function of genes encoding carbohydrate active enzymes: some of the glycoside hydrolases (GH28, polygalacturonase) and polysaccharide lyases (PL4, rhamnogalacturonate lyase; PL22, oligogalacturonate lyase) are fairly well characterized and have specific functions in the depolymerization of pectin polysaccharides (see also the references in our manuscript). Additional information including the respective references can be found in the CAZy database (doi 10.1093/nar/gkt1178). To further support the polysaccharide degradation potential of MAG CO124, we performed an additional experiment (please see the additional figure for review purpose provided in response to Reviewer 1 above). We cultivated *Bryobacter aggregatus*, the closest cultured relative that has been described in detail, either with pectin or with glucose under oxic conditions. *B. aggregatus* can utilize pectin (doi 10.1099/ijms.0.013250-0) although a much lower number of glycoside hydrolases and polysaccharide lyases were encoded in its genome compared to MAG CO124 (Supplementary Figure S5). This also includes the well characterized, pectin-specific families GH28 and PL22. In the transcriptomes sequenced from our *B. aggregatus* cultures, several of the pectin-specific carbohydrate active enzymes were upregulated or only active when grown on pectin compared to the cultures grown on glucose (additional figure for review purpose, see answers to reviewer 1). Similar to MAG CO124, *B. aggregatus* transcribed a polygalacturonase, oligogalacturonate lyase and putative rhamnogalacturonan lyases during growth with pectin as carbon and energy source. This strongly supports the suggested role of the carbohydrate active enzymes, with the encoding genes being transcribed or even transcriptionally upregulated in MAG CO124 under anoxic/sulfate-reducing conditions in our bioreactor setup.

Please find below our point-by-point response to your minor concerns.

I also had a small number of minor concerns:

396: Scaffold coverage was obtained by mapping with Bowtie2, but how was coverage derived from the mapping results?

Response: This has been stated in more detail now: "... coverage information was generated using the script `jgi_summarize_bam_contig_depths` included in MetaBAT2 (version 2.12.1)", (L 422-423).

P-values are reported for the metatranscriptomics, but these are derived from 2 conditions with 4 technical replicates each. It would be clearer for the reader if this was stated in the figure legend.

Response: The legends of Fig. 3 and Fig. 4 have been changed accordingly.

225: Ref 16 is a study of peat.

Response: Thank you. This has been changed.

268: Most abundant constituents of what?

Response: The most abundant constituents of pectin. This has been added to the sentence.

266 onwards: Reading the manuscript, it was unclear what analyses these conclusions are based on, at this stage in the manuscript. Also, since they are purely bioinformatic, they were quite definitive given the heterogeneity of functions (and simple lack of characterisation) found in many GH families. The conclusions should be stated less confidently.

Response: Thanks for pointing that out. As mentioned above, some of the carbohydrate active enzyme families are quite well characterized and have very specific functions in the degradation of pectin polysaccharides, e.g., GH28 (polygalacturonase), PL4 (rhamnogalacturonate lyase) and PL22 (oligogalacturonate lyase). The additional experiment that we performed (see above) strongly supports the suggested role of the carbohydrate active enzymes transcribed in MAG CO124 (additional figure for review purpose).

Reviewer #1 (Remarks to the Author):

The authors have answered my comments mostly in a satisfactory way. Additional data were collected that clarified the abundance of the sulfate-reducing Acidobacteria target bacterium: It is of minor relevance (0.1%) in the bioreactor operated for this study. The authors argue that this is not important for defining the ecophysiology of these "new" sulfate-reducing Acidobacteria – and they have a point in saying so. Nevertheless, the direct uptake of carbohydrates into their new strain was not demonstrated, albeit transcriptomic levels for sugar utilization were indicated. Still there is the grain of salt that many readers of Nature journals want to know certainly whether such a new sulfate-reducing bacterium on the block also has relevance in the turnover of carbohydrates in the habitat from which it was enriched.

Reviewer #2 (Remarks to the Author):

I am satisfied with the answers to the previously addressed points, the extra experiments and the additional discussion in the manuscript clarifies better the metabolism of Acidobacteria

Reviewer #3 (Remarks to the Author):

The authors have addressed the concerns raised in my initial review.

Response: We are grateful for the encouraging feedback and that all three reviewers were satisfied by our additional experiments and responses. We want to thank all reviewers for the time and effort they dedicated in reviewing our manuscript.

REVIEWERS' COMMENTS

Reviewer #1 (Remarks to the Author):

The authors have answered my comments mostly in a satisfactory way. Additional data were collected that clarified the abundance of the sulfate-reducing Acidobacteria target bacterium: It is of minor relevance (0.1%) in the bioreactor operated for this study. The authors argue that this is not important for defining the ecophysiology of these “new” sulfate-reducing Acidobacteria – and they have a point in saying so. Nevertheless, the direct uptake of carbohydrates into their new strain was not demonstrated, albeit transcriptomic levels for sugar utilization were indicated. Still there is the grain of salt that many readers of Nature journals want to know certainly whether such a new sulfate-reducing bacterium on the block also has relevance in the turnover of carbohydrates in the habitat from which it was enriched.

Response: We are glad that Reviewer 1 is now satisfied by the additional experiments provided. As pointed out in our previous response, we cultivated *Bryobacter aggregatus* strain MLP3 as the closest cultured relative of Acidobacterium MAG CO124 with either pectin or glucose. We observed a clear upregulation of pectin-specific carbohydrate-active enzymes in the transcriptome of *B. aggregatus* MLP3 when grown on pectin as compared to growth on glucose. This supports the suggested role of MAG CO124's glycoside hydrolases and polysaccharide lyases (please see the figure provided for review purposes in revision 1).

Reviewer #2 (Remarks to the Author):

I am satisfied with the answers to the previously addressed points, the extra experiments and the additional discussion in the manuscript clarifies better the metabolism of Acidobacteria

Response: Thank you very much for your positive feedback.

Reviewer #3 (Remarks to the Author):

The authors have addressed the concerns raised in my initial review.

Response: We very much appreciate your supportive feedback.